# A bistable prokaryotic differentiation system underlying development of conjugative transfer competence

**Sandra Sulser**[‡], **Andrea Vucicevic**[‡], **Veronica Bellini, Roxane Moritz, François Delavat**[¤], **Vladimir Sentchilo, Nicolas Carraro, Jan Roelof van der Meer**[*]

Department of Fundamental Microbiology, University of Lausanne, Lausanne, Switzerland

¤ Current address: Nantes Université, CNRS, US2B, UMR6286, Nantes, France
‡ should both be considered first author
* Janroelof.vandermeer@unil.ch

**Data Availability Statement:** The sequence read data belonging to the P. putida ICE transcriptomes are available from the Short Read Archives under project number PRJNA784540 All further relevant data are within the manuscript and its Supporting

## Abstract

The mechanisms and impact of horizontal gene transfer processes to distribute gene functions with potential adaptive benefit among prokaryotes have been well documented. In contrast, little is known about the life-style of mobile elements mediating horizontal gene transfer, whereas this is the ultimate determinant for their transfer fitness. Here, we investigate the life-style of an integrative and conjugative element (ICE) within the genus *Pseudomonas* that is a model for a widespread family transmitting genes for xenobiotic compound metabolism and antibiotic resistances. Previous work showed bimodal ICE activation, but by using single cell time-lapse microscopy coupled to combinations of chromosomally integrated single copy ICE promoter-driven fluorescence reporters, RNA sequencing and mutant analysis, we now describe the complete regulon leading to the arisal of differentiated dedicated transfer competent cells. The regulon encompasses at least three regulatory nodes and five (possibly six) further conserved gene clusters on the ICE that all become expressed under stationary phase conditions. Time-lapse microscopy indicated expression of two regulatory nodes (i.e., *bisR* and *alpA-bisDC*) to precede that of the other clusters. Notably, expression of all clusters except of *bisR* was confined to the same cell subpopulation, and was dependent on the same key ICE regulatory factors. The ICE thus only transfers from a small fraction of cells in a population, with an estimated proportion of between 1.7–4%, which express various components of a dedicated transfer competence program imposed by the ICE, and form the centerpiece of ICE conjugation. The components mediating transfer competence are widely conserved, underscoring their selected fitness for efficient transfer of this class of mobile elements.

## Author summary

Horizontal gene transfer processes among prokaryotes have raised wide interest, which is attested by broad public health concern of rapid spread of antibiotic resistances. However, we typically take for granted that horizontal transfer is the result of some underlying

information files. Source data for all figures are provided.

**Funding:** This research has been supported by Swiss National Science Foundation (https://snf.ch/en) grants 31003B_156926/1, 31003A_175638 and 310030_204897 to JvdM. The funders had no role in study design, data collection and analysis, decision to publish or preparation of the manuscript.

**Competing interests:** The authors have declared that no competing interests exist.

spontaneous low frequency event, but this is not necessarily the case. As we show here, mobile genetic elements from the class of integrative and conjugative elements (ICEs) impose a coordinated program on the host cell in order to transfer, leading to an exclusive differentiated set of transfer competent cells. We base our conclusions on single cell microscopy studies to compare the rare activation of ICE promoters in individual cells in bacterial populations, and on mutant and RNA-seq analysis to show their dependency on ICE factors. This is an important finding because it implies that conjugation itself is subject to natural selection, which would lead to selection of fitter elements that transfer better or become more widespread.

## Introduction

Prokaryote genomes typically contain a variety of mobile genetic elements (MGEs), such as plasmids [1], phages, transposons or integrative and conjugative elements (ICEs) [2,3], which largely contribute to host evolution and community-wide adaptation through genome rearrangements and horizontal gene transfer [4–6]. Although transfer mechanisms per se are well understood, it is insufficiently appreciated that MGEs form their own entities, which are embedded within a host, but undergo selection towards their own fitness optimization [7–11]; for example, by increasing transfer success to new cells [12,13]. MGE decisions potentially oppose the interests of the host cell [14,15] and can inflict serious damage by cell lysis [16,17] or cell division inhibition [18–20]. In order to operate independently, MGE regulatory networks need specific components to pursue their own program, while impinging on other host factors and signals. Apart from bacteriophage development [21,22], it is mostly unknown how MGEs operate within the host regulatory system [23]. In order to study fitness selection and regulatory control in horizontal gene transfer, we focus here on a class of mobile DNA elements called ICEs. Transfer of ICEs is characterized by a transition from a chromosomally integrated to an excised state with a circularized ICE-DNA that can conjugate from the host cell to a new recipient (Fig 1A) [24]. ICEs come in various families with distinct and mosaic evolutionary origins [24–26], which operate different regulatory mechanisms to control the excision transition state [27]. The *clc* integrative and conjugative element in *Pseudomonas* (ICE*clc*) that we use here, is a model for a widespread family occurring in opportunistic bacteria including pathogenic *Pseudomonas aeruginosa* [28,29]. ICE*clc*-type elements have been implicated in transmission of antibiotic resistances [30–32] and xenobiotic metabolism [28,33], lending broad significance for understanding the molecular and regulatory basis of their evolutionary success.

ICE*clc* is 103 kb in size and present in two integrated identical copies in the bacterium *P. knackmussii* strain B13 [28], an organism isolated for its capacity to grow on the xenobiotic compound 3-chlorobenzoate (3CBA) [34]. Characteristic for ICE*clc* is activation of the promoter of the ICE*clc* integrase ($P_{int}$) [35] and of the *integrase regulatory factor* gene *inrR* [36] in a subpopulation of cells in stationary phase after growth on 3CBA as sole added carbon source [37]. This was judged from single cell studies of *P. knackmussii* or *Pseudomonas putida* with integrated ICE*clc* equipped with single-copy promoter-fluorescent reporter gene fusions. ICE*clc* excision [13], temporary replication [12] and transfer [13] are only observed from such activated cells, suggesting they (and only they) are responsible for ICE transfer (Fig 1A). If so, this would imply that ICE*clc* is capable to initiate and orchestrate a specific program in a subset of host cells that makes them competent for transfer. The nature of this transfer competence program, its temporal coordination in individual cells and the mechanisms of its

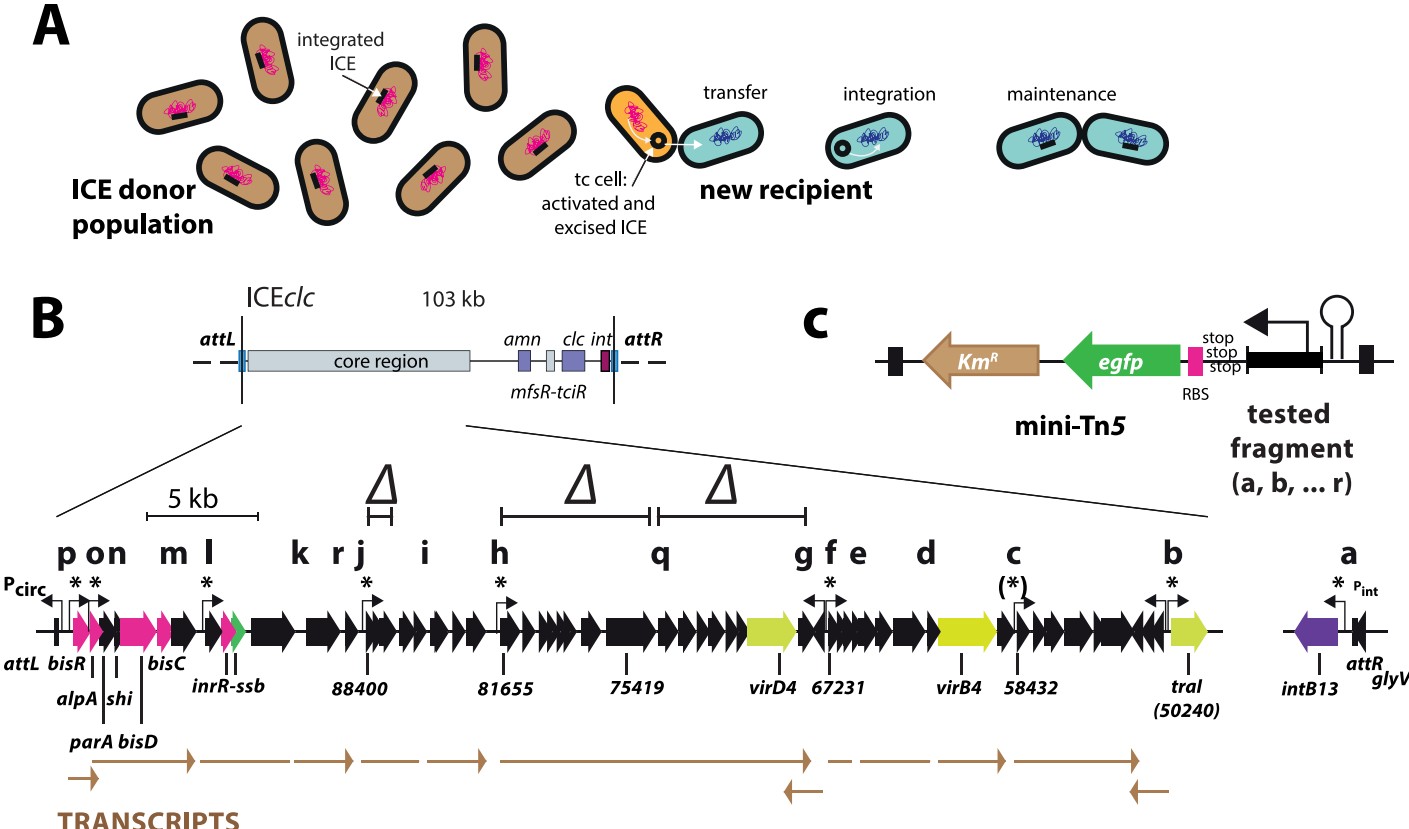

**Fig 1. ICE*clc* life-style and transfer competence development. A** Chromosomally-integrated ICE (black bar, schematically) activates and excises in rare transfer competent cells (tc, black circle; cell highlighted in orange), and transfers by conjugation to a new recipient (here in blue shading), where it inserts site-specifically and maintains through chromosomal replication. **B** Integrated ICE*clc* is delineated between the two attachment (*att*) sites (vertical lines). Layout shows the location of the core region relative to the integrase gene, the variable gene region with the *clc* genes for chlorocatechol and *amn* for 2-aminophenol metabolism, and the key regulator genes *mfsR-tciR*. Gene map below shows individual core genes (black and colored arrows, relevant gene names underneath), previously Northern-mapped transcripts and their orientation (in brown), fragments tested for promoter studies (letters on top) and hooked arrows pointing to identified promoters ($P_{58432}$ being inconclusive). Asterisks point to those promoters being expressed in tc cell subpopulations. Regions indicated with a Δ denote deletions for transfer studies. **C** General strategy of single copy chromosomally delivered individual or paired promoter-fluorescent gene reporter fusions using mini-transposon delivery.

restriction to this subpopulation of tc cells, have remained elusive and are the main focus of this work.

ICE*clc* transfer competence is most likely encoded by a core set of ~40 genes (Fig 1B) that is highly conserved among (presumed) integrated ICEs in a wide variety of *Proteobacteria* [28,33]. To our current understanding, ICE*clc* remains 'silent' in dividing cells through repression by a protein named MfsR [38] (Fig 2A). MfsR repression limits formation a second regulatory protein named TciR [38], which itself is a transcriptional activator for a recently discovered third regulator, named BisR [29] (Fig 2A). BisR activates a promoter upstream of *alpA*, which initiates a self-regulatory feedback loop that is maintained by a transcription activator complex named BisDC [29] (Fig 2A). Modeling and experimental observations on a reduced gene set including *bisR* and *bisDC* suggested that this feedback loop can generate an ON/OFF downstream response [29]. We hypothesized that cells with an 'ON' feedback loop follow a state leading to ICE excision and transfer, whereas the ICE remains silent in cells with a dissipated 'OFF' loop. This resembles a 'bistability' generator in which the ICE can follow either of two paths that are mutually exclusive: remain integrated and silent, or be activated

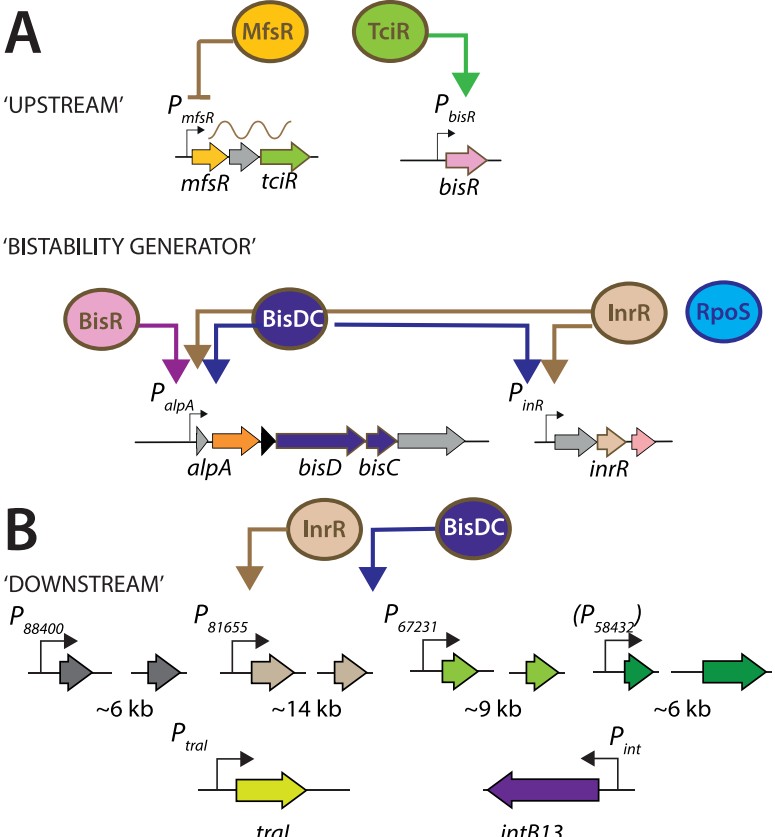

**Fig 2. Proposed ICE*clc* transfer competence regulon. A** Current state of knowledge of the 'upstream' factors MfsR, TciR and BisR, acting sequentially. The 'bistability generator' feedback loop initiated by BisR, but maintained by BisDC. RpoS being optimal for *inrR* expression. **B** Identified 'downstream' targets of the transfer competence regulon and their control by BisDC and InrR. Promoters from *alpA* onwards activated in the same tc cell subpopulation.

and prepare for transfer (Fig 1A). If this hypothesis were valid, we would expect the different genes encompassing the transfer competence program to be expressed in essentially the same subpopulation of cells. In order to test this, we studied subpopulation expression from single-copy chromosomally integrated promoter fusions to fluorescent reporter genes in *P. knackmussii* (with its two identical ICE*clc* copies), and in some cases for technical reasons, in *P. putida* with a single integrated ICE*clc*. Since the downstream parts of the ICE*clc* transfer competence program (except for the integrase gene) were not known at the study onset, we selected all potential promoter regions from transcriptional units within the conserved ICE*clc* core gene region that were previously described by Northern studies [39]. Promoters were tested alone and in paired combinations with P$_{int}$ to study their temporal expression differences in cell subpopulations by time-lapse imaging. We verified the dependency of identified promoters on key ICE*clc* regulatory elements, using single-cell fluorescence imaging and RNA-seq, and studied functionality and conservation of a number of identified downstream gene regions (Fig 2B) on ICE transfer. Our results indicate a multi gene cluster network of ICE*clc* transfer competence formation ('regulon') that despite temporal noisiness is followed by and restricted to a subset of cells, ensuring streamlined ICE*clc* transfer.

## Results

### Identification of subpopulation-expressed promoters within the conserved core region of ICE*clc*

To identify the gene regions possibly implicated in the ICE*clc* transfer competence regulon, we inspected all the putative promoters in the conserved core region of ICE*clc* (Fig 1B). Putative promoter regions were selected based on a previously conducted Northern transcript analysis (Fig 1B, brown arrows) [39]. Eighteen fragments were amplified by PCR, fused to a promoter-less *egfp* gene with its own ribosome binding site (Figs 1C and S1), and placed in single copy on the chromosome of *P. knackmussii* or *P. putida* with integrated ICE*clc*. Fluorescent protein expression was examined in stationary phase cultures grown on 3CBA for three clones of each fusion construct, inserted at different random chromosomal positions, in order to control for single-copy positional effects. We looked specifically at expression in subpopulations, as this would be a hallmark for ICE*clc* activation.

Nine cloned regions yielded clear eGFP expression in stationary phase cells, with a typical small proportion of brighter cells amidst the rest (Fig 3A). Their bimodal expression becomes more apparent from quantile-quantile (qq-plot) analysis, yielding two separate population distributions, the largest of which with low baseline expression and the smaller with distinct higher eGFP expression (white and yellow zones, respectively, Fig 3B and S1 Data). This contrasts to the eGFP fluorescence distributions observed with the other tested regions, which did not show any deviation from a single expected unimodal distribution (Fig 3C). To confirm the significance of this, we repeated the subpopulation analysis for three independent clones with different single-copy promoter insertion positions that were expected to vary slightly. Determined subpopulation averages were significantly higher for the nine promoter regions than the mean eGFP fluorescence of three independently determined main populations (compare magenta to brown bars, Fig 3D, $n = 3$ replicates, p-values from paired t-tests). For none of the other fragments in any of the three clones tested, subpopulations of cells with higher eGFP expression were apparent (Fig 3D). Some differences in the mean fluorescence levels of their main cell populations were visible, some being higher (i.e., UR-parA, UR97571, UR89247 and UR73676) than a background control of a strain carrying a deletion in a subpopulation-dependent expressed promoter (i.e., 50240Δ, Fig 3C and 3D). Others were no different from background (e.g., UR89746, UR84835, UR66202 and UR62755). These results suggested that nine regions potentially comprise ICE*clc* promoters with bimodal behaviour (highlighted with asterisks in Fig 3D), whereas the other nine do not. Global transcriptome analysis by reverse-transcribed RNA sequencing (RNA-seq, S2 Fig) suggested some of the latter to have weak but population-wide promoter activity (e.g., the UR67800-region). The expressed eGFP reporter intensity varied among the nine 'bimodal' promoters (Fig 3B and 3D), which may be a consequence of their local architecture and sequence differences. Eight of the nine promoters (except $P_{bisR}$) share a common sequence motif that may point to shared regulatory factors (S3 Fig; see further below). The average proportion of the stationary phase subpopulation with higher eGFP expression (defined by the qq-plot threshold and visualized by the yellow zone in Fig 3B) varied between 1.5 and 7.5% (Table 1). The highest subpopulation of cells was detected for the *bisR* promoter, which expressed in 9.1% of cells (Table 1).

Inspection of read coverages from RNA-seq of *P. putida* cells carrying ICE*clc* in exponential growth on 3CBA and subsequent stationary phase, and in stationary phase of succinate-grown cells broadly confirmed specific transcription initiation of the suspected promoter regions (S2 Fig). Read coverages in the upstream regions of the open reading frames (ORFs) 67231, 81655, 88400, *alpA* and *inrR* clearly and strongly increased in stationary phase compared to exponential growth on 3CBA, and in 3CBA versus succinate stationary phase (S2 Fig). In contrast, the

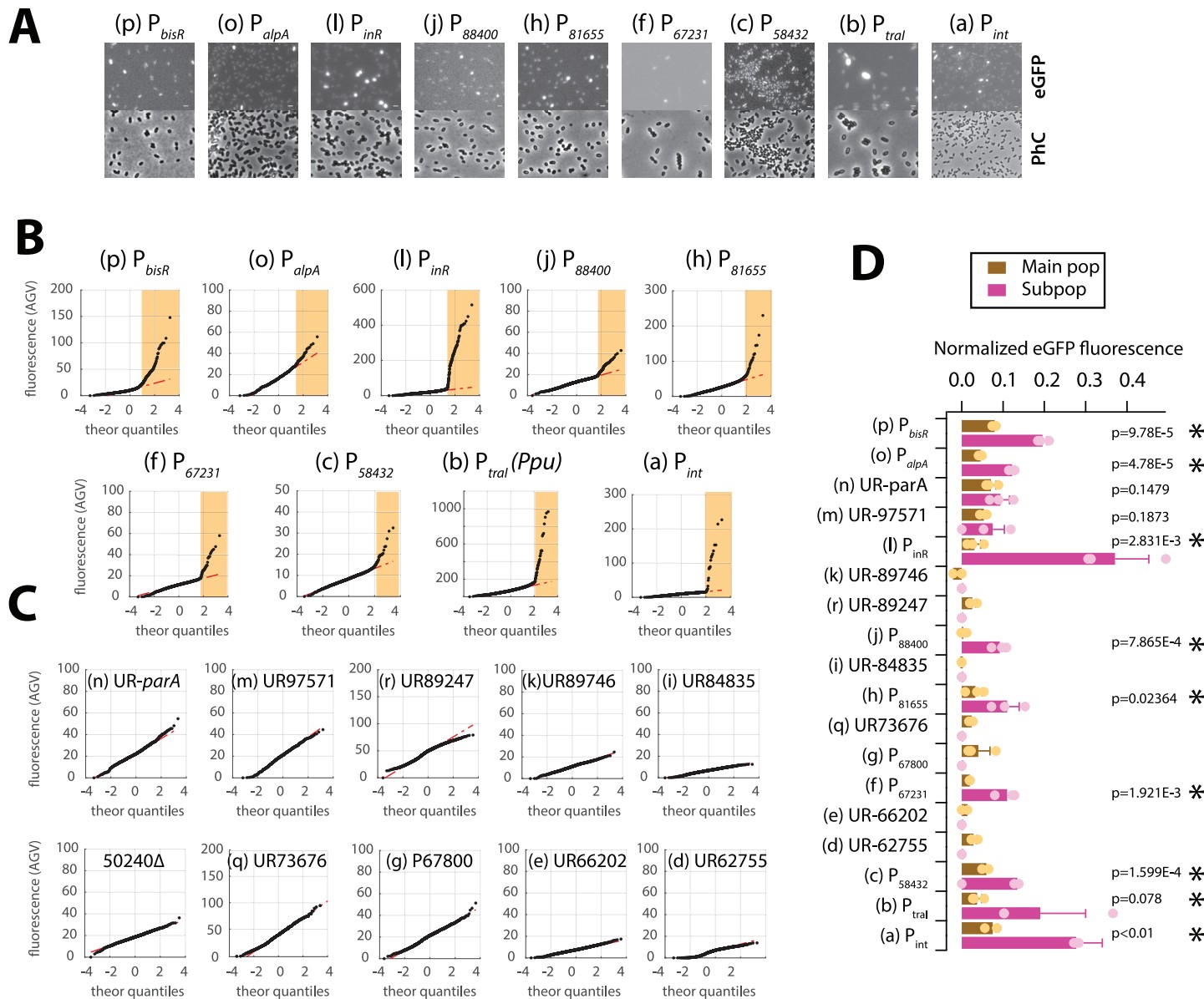

**Fig 3. Identification of tc cell subpopulation-specific ICE*clc* promoters. A** Auto-contrasted and cropped micrographs of *P. knackmussii* B13 strains (exception construct b in *P. putida* ICE*clc*, Ppu) grown on 3CBA imaged in stationary phase for eGFP fluorescence of the indicated construct or in phase-contrast (PhC). White bars indicate 1 μm length. Letters and fragments corresponding to Fig 1 locations. P, putative promoter; UR, upstream region. **B** Quantile-quantile plot representations of stationary phase reporter fluorescence (each dot is value from a single cell; red line is linear regression on the fluorescence distribution of the main population). Yellow zones point to tc cell subpopulations. Fluorescence values scaled to zero by subtracting image background. **C** As (**B**) but for the tested constructs without detectable subpopulation expression. **D** Mean normalized fluorescence values (bars) ± one *SD* (whiskers) from 3 replicate strains of *P. knackmussii* B13 or *P. putida* ICE*clc* (construct b) in stationary phase on 3CBA with independent mini-transposon-inserted reporter constructs. Brown bars, value of the main population; magenta bars, values of the identified tc cell subpopulation (if any). Colored dots, individual replicate values. Asterisks point to those constructs showing significantly higher promoter expression in the subpopulation than in the main population (*n* = 3 or 4, p-values from paired one-sided t-test).

region upstream of ORF58432 did not show any depletion of mapped reads (S2 Fig), and, thus, does not confirm the weak observed promoter activity in isolation (Fig 3B). On the other hand, upstream regions of the ORFs 73676, 84835 and 89247 showed clear increase in mapped read coverage in 3CBA-grown stationary phase compared to exponential phase or succinate-stationary phase cultures (S2 Fig), but this was not associated with stationary phase expression

**Table 1. Proportion of stationary phase *P. knackmussii* B13 or *P. putida* ICE*clc* cells expressing eGFP fluorescence from single-copy integrated bistable ICE*clc* promoter reporter fusions.**

| Promoter region | Wild-type ICE*clc*[a] | Δ*mfsR*[b] |
|---|---|---|
| *bisR* | 9.10 ± 0.87 | 100 |
| *alpA* | 3.10 ± 1.07 | 49.4 ± 2.5 |
| *inrR* | 7.51 ± 1.78 | 77.1 ± 22.0 |
| *orf88400* | 2.30 ± 0.13 | 16.4 ± 2.0 |
| *orf81655* | 1.47 ± 0.47 | 42.2 ± 1.5 |
| *orf67231* | 7.03 ± 2.22 | 40.1 ± 12.3 |
| *orf58432* | 1.59 ± 0.37 | ND[d] |
| *traI* | 2.21 ± 0.70[c] | ND |
| *intB13* | 2.92 ± 0.6[e] | 79.0 ± 17.9 |

[a] Mean proportion ± one SD of *P. knackmussii* B13 cells expressing eGFP from the corresponding promoter reporter fusion, after 72 h of growth in MM with 5 mM 3CBA, determined by quantile-quantile-plotting (n = 3 independent clones).

[b] As a, but in *P. putida* UWC1 ICE*clc*-Δ*mfsR*. Note that quantile-quantile plotting is not very accurate to determine large subpopulation size fractions [42].

[c] As a, but in *P. putida* UWC1 ICE*clc*.

[d] ND, not determined.

[e] Raw image data reanalyzed from ref [42].

of single-copy ectopically placed fragments transcriptionally fused to promoterless *egfp* (Fig 3D). We cannot exclude that these cloned fragments were too small or otherwise did not encompass a complete promoter region. Alternatively, the observed changes in read coverage in these three regions may have been due to transcript processing as previously suspected [39]. RNA-seq did not show any major depletion of mapped reads for the regions UR62755, UR66202 and UR97571 (S2 Fig), suggesting no promoter presence and thus confirming single cell reporter results (Fig 3C). None of the other ICE*clc* regions showed the presence of the conserved sequence motif found in eight of nine bimodal expressed promoters (S3 Fig). Mapped read abundances from RNA-seq varied considerably among ICE*clc* core transcripts, but did not necessarily correlate to the measured eGFP fluorescence level from cloned fragments in single cells. For example, the read coverage upstream of ORF81655 was the highest of all, whereas that upstream of the *intB13* gene was much lower (S2 Fig). In contrast, eGFP expression from both single copy $P_{int}$-*egfp* and $P_{81655}$-*egfp* fusions was similar (Fig 3B), which suggests additional post-transcriptional regulatory mechanisms to act on them.

It should be noted that RNA-seq captures an average from all cells in culture and, therefore, does not exclusively quantify transcripts in subpopulations of transfer competent cells. Coverage plots and mapped read directions also suggested specific transcription from the ORFs oriented oppositely to *traI* (ORF52324), from ORF67800 and for the previously mapped $P_{circ}$-promoter [40], indicative for promoters that would be independent from 3CBA stationary phase conditions (S2 Fig).

Collectively, these results indicated that a total of nine upstream regions of the ICE*clc* core genes showed bimodal expression in a subpopulation of stationary phase cells after growth on 3CBA. One of these ($P_{58432}$) is either weak or does not comprise an independent promoter, even though it carries a promoter motif common to the others (S3 Fig). This suggested they may be part of the same transfer competent regulon.

## Bimodally expressing promoters and P_int expression colocalize in the same subpopulations of cells

In order to further determine whether the nine identified bimodally expressed ICE*clc* promoters might belong to a regulon operating in the same individual cells, we compared their expression from single-copy fluorescent reporter insertions with that of a single-copy $P_{int}$-*mcherry* fusion in the same cell (placed in the same chromosomal position for all *P. knackmussii* strain comparisons). Stationary phase expression in 3CBA-grown cultures was clearly correlated to that of $P_{int}$ in the case of the $P_{alpA}$, $P_{inR}$, $P_{88400}$, $P_{81655}$, $P_{67231}$ and $P_{traI}$ promoters, but not in case of $P_{bisR}$ (Fig 4A–4H, 13.8% overlap of $P_{bisR}$-*egfp* and $P_{int}$-*mcherry* expression in the identified subpopulations, as opposed to 54.8–91.1% overlap for the others; S2 Data). Between

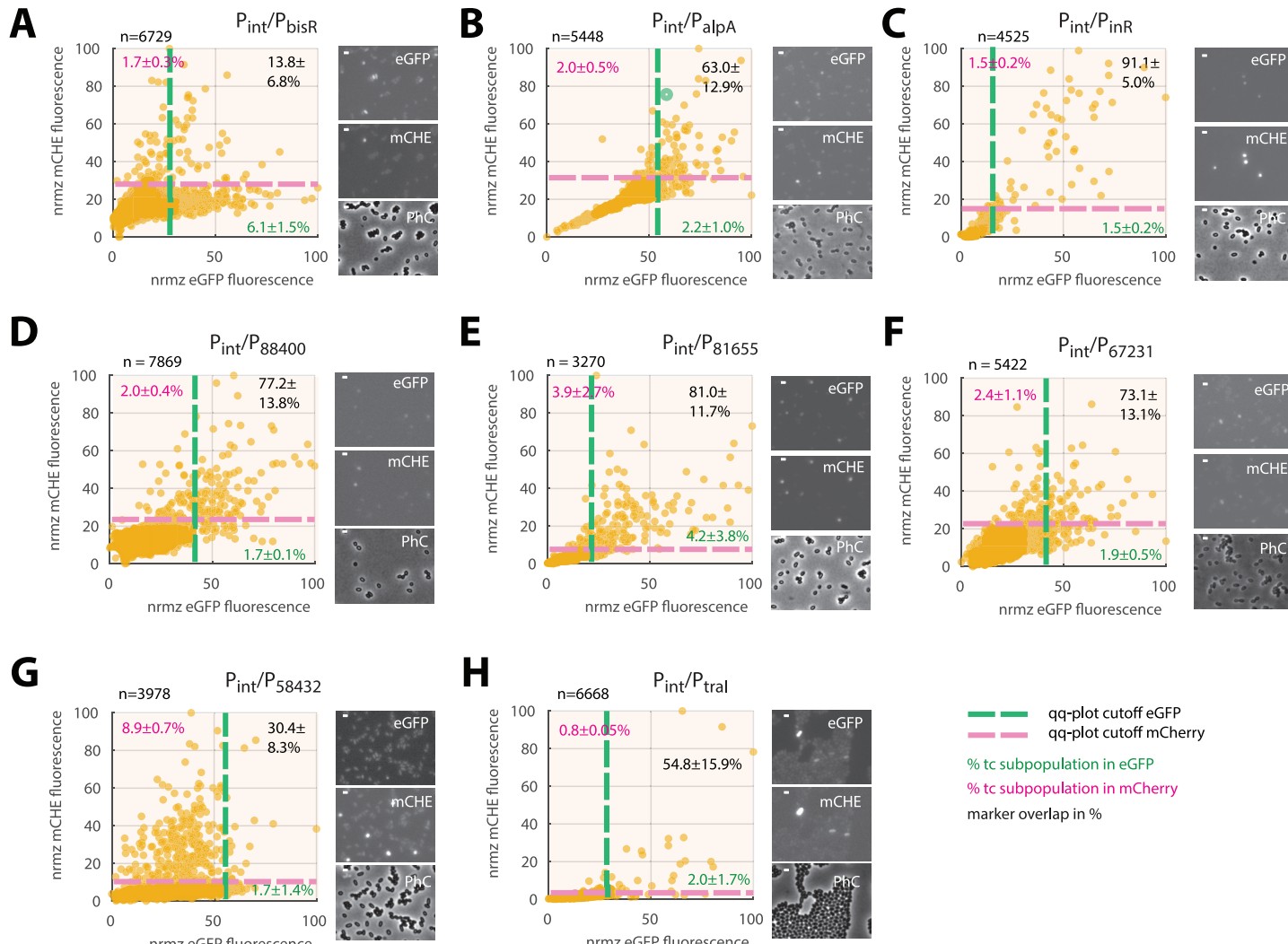

**Fig 4. Colocalization of expression of paired bimodal promoters from the ICE*clc* transfer competence regulon.** Dots show stationary phase normalized fluorescence values (72-h-cultures on 3CBA, background subtracted and scaled to the maximum value within a data set) of individual *P. knackmussii* B13 (**A-G**) or *P. putida* ICE*clc* (**H**) cells carrying the indicated (single copy integrated) double reporter construct. Insertion position of $P_{int}$-*mcherry* the same in all strains, except for **H** (*P. putida*). Plots combined from three biological replicates (*n* is total amount of analyzed cells) with micrographs on the right showing example images in GFP, mCherry and phase-contrast (auto-contrasted for display). Dashed magenta and green lines delineate the main from subpopulations (defined from quantile-quantile plots as in Fig 3B, percentages showing the mean subpopulation size ± one *SD* from *n* = 3 biological replicates). Percentages on the upper right indicate the mean proportion ± one *sd* of cells in the overlap (brown) of green and magenta channels. Bars within micrographs indicate 1 μm.

0.8–4.2% of cells imaged at a single stationary phase time point classified as being part of the higher-expressing subpopulation (Fig 4B–4H, based on qq-plotting), which was larger for P$_{bisR}$-eGFP-expressing cells (Fig 4A, 6.1%), similar as noted before for the individual promoter fusions (Table 1). Of note, the combination of P$_{int}$ with P$_{traI}$-*egfp* was tested in *P. putida* ICE*clc*, because we did not manage to obtain this genetic construct properly in *P. knackmussii* B13. Marker correlation was less clear for the combination of P$_{int}$ with P$_{58432}$-*egfp* (Fig 4G, 30.4% of marker overlap), which might be due to its inherently low expression (e.g., Fig 2B) and higher uncertainty to define subpopulations of cells with aberrant expression (see below). These results thus indicated that 7 of 9 bimodal ICE-promoters express in the same individual cells, whereas two-thirds of cells expressing *bisR* do not express P$_{int}$.

To understand how differences in temporal expression would influence observed marker correlations in cell subpopulations, we next quantified dynamic reporter fusion expression in growing and stationary phase cells for the double-labeled strains (Fig 5 and S3 Data). Hereto, cells were seeded on agarose surfaces with 3CBA and automatically imaged at 4–10 replicate positions every 30 min (to avoid fluorescence bleaching). Cells went through on average roughly four cell divisions before reaching stationary phase, at the onset of which or slightly afterwards, reporter fluorescence from ICE-promoters became evident (Fig 5A; blue dotted lines show population growth; solid colored lines show single cell paired fluorescence). Individual cells with higher than unimodally expected marker fluorescence continued to appear throughout stationary phase, although their appearance rate 'peaked' during a time window of some 20 h (Fig 5B). Despite *P. knackmussii* strains having the same insertion position of the P$_{int}$-*mcherry* marker, mCherry fluorescence itself varied substantially as a function of the second (randomly inserted) reporter construct (Fig 5A). Furthermore, expression varied visibly among individual cells, both in the intensity of the fluorescence and its timing. This is suggestive for extreme noisiness that may be the consequence of low (fluctuating) numbers of relevant transcription factors. As before, eGFP expression from the P$_{58432}$-promoter in combination with P$_{int}$-*mcherry* was weak, and did not show any signs of temporal increase in individual cells as for the other promoter fusions (Fig 5A). This thus indicates that this fragment in combination is not active as independent promoter.

To determine the onset of promoter expression, we quantified at every time point by qq-thresholding the proportion of cells showing higher than unimodally expected reporter fluorescence, and compared this among strains from their common P$_{int}$-*mcherry* marker (Fig 5B). Seen across the entire subpopulation, both *bisR* and *alpA* promoters appeared to become active some 10 h before P$_{int}$, whereas the others showed the same temporal activation as P$_{int}$ (Figs 5B and S4). Expression of *bisR* was also clearly more abundant (~200 identified cells) than that of P$_{int}$ (30 cells) across the same set of observed cells and using the same qq-plot criteria to distinguish cells with higher than expected fluorescence (Fig 5B). Visualized as pairs across individual cells, the *bisR* and *alpA* promoters also fired before the P$_{int}$-promoter (Fig 5C, blue dashed slope lines), whereas they were similar for P$_{int}$ and the P$_{67231}$ or P$_{inR}$-promoters (Fig 5C, yellow dashed slopes), and the others (S4 Fig). Despite noisiness in reporter expression in individual cells, which resembled that of the partial overlap in the scatter plots of Fig 4 (i.e., only one of both markers expressed above the qq-plot threshold), correlated temporal expression was clearly visible in those cells where both reporter markers were above qq-thresholds. In summary, the time-lapse imaging thus indicated that expression of the bimodal ICE core promoters is correlated in the same individual cells and with roughly the same temporal dynamics, which is evidence for them being part of the same transfer competence regulon. Activation of *bisR* and *alpA* promoters occurs earlier, probably because, as we will further discuss below, they comprise key 'upstream' regulatory nodes in the transfer competence pathway of ICE*clc* (Fig 2) [29]. The fact that two-thirds of cells with eGFP fluorescence from the *bisR*-promoter

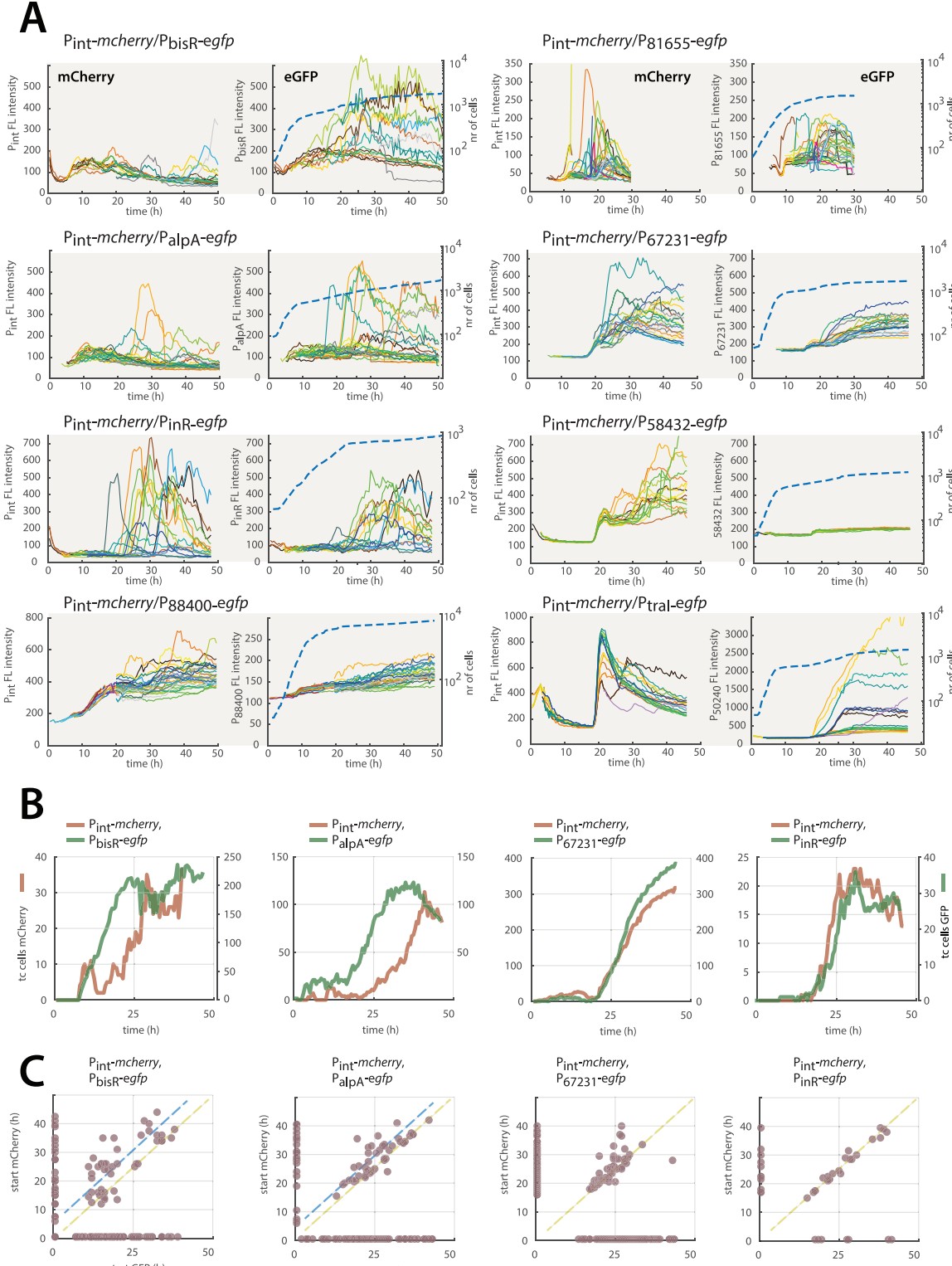

**Fig 5. Temporal expression of paired promoters from the ICE*clc* transfer competence regulon. A** Time-lapse fluorescence of selected identified tc cells of surface-grown *P. knackmussii* B13 with indicated single-copy inserted promoter-fluorescent reporter constructs; each line corresponding to an individual cell traced over time with corresponding colors between fluorescence channels. Thick blue dashed line indicates population growth (scale on the right) for the analyzed shown image area. Fluorescence values shown here as direct camera greyscale output, averaged over each individual cell area. The $P_{int}$-$P_{88400}$ pair was manually stitched at t = 19 h

because of image drift in time-lapse; that of $P_{int}$-$P_{81655}$ stopped at t = 30 h because of loss of automated focus. $P_{int}$-$P_{traI}$ is in *P. putida* ICE*clc* host background. **B** Time increase of the identified subpopulation sizes from each reporter fluorescence individually (as being above a floating qq-plot threshold at each individual time point). Note how the inferred tc cell populations from eGFP fluorescence of the *bisR* and *alpA* promoter fusions increase before that of mCherry from $P_{int}$, but not in case of eGFP from the $P_{67231}$ or $P_{inR}$-promoters. Also note how six times more cells express eGFP from the *bisR* promoter than mCherry from $P_{int}$. **C** Correlation of paired fluorescent reporter expression from single-copy inserted constructs of the indicated promoter pairs in identified *P. knackmussii* tc cells (on either of the two fluorescence markers). Dots show the derived start of the increase of the fluorescence signal (as in trace lines of panel **A**). In case no increase above background fluctuation could be measured (i.e., slope less than 8 fluorescence units over five consecutive time points, and $r^2$ below 0.9800), the value was arbitrarily set at 1 to allow plotting. *intB13* promoter in all cases coupled to mCherry and in the same insertion position in *P. knackmussii*. Yellow dotted line indicates exact matching of starts; blue dotted lines indicate deviation from matching starts (eGFP expressing earlier than mCherry). Remaining promoter pairs shown in S4 Fig.

do not subsequently activate the 'downstream' $P_{int}$-promoter suggests that *bisR*-expression is necessary but not sufficient to continue with full transfer competence.

## Dependency of core promoter expression on known ICE*clc* regulatory factors

In order to provide further evidence for the identified ICE*clc* promoters being part of the same transfer competence regulon, we examined their expression in dependency of previously identified key regulatory elements for activation of ICE*clc*: *mfsR* [38], *inrR* [36], *bisR* and *bisDC* [29].

The relative normalized abundance of mapped transcripts from RNA-seq in the ICE*clc* conserved core region strongly increased in stationary phase *P. putida* ICE*clc* background with a deletion in *mfsR* (Fig 6A, Δ*mfsR* STAT; S4 Data). MfsR is the major global transcription repressor of ICE*clc* activation [38], plus that of a set of genes on ICE*clc* coding for an efflux pump [41]. Indeed, *mfsR* deletion resulted in increased expression of *tciR* (Fig 6A and 6B), and of the multidrug efflux pump genes (*mfsABC*, Fig 6A) [41]. Deregulated *tciR* also led to higher *bisR* expression (Fig 6A, EXPO), which is linked to ICE*clc* activation [29]. Global transcript abundances of the ICE*clc* core genes were 2–32 fold higher in stationary phase 3CBA-grown cells of *P. putida* ICE*clc*-Δ*mfsR* than wild-type ICE*clc* (Fig 6B). Expression of ORF52324 and ORF67800 was unaffected, confirming they are not part of the bistability regulon (Fig 6A and 6B). The increased globally observed expression of the ICE core genes in the Δ*mfsR* strain was primarily due to a sharp increase of the proportion of cells activating the ICE compared to wild-type ICE*clc* (Table 1). When additionally to *mfsR* also the *bisR* gene was deleted, stationary phase expression of the ICE*clc* core genes was strongly reduced, even lower than in wild-type (Fig 6A and 6B). The reason for this is that in absence of BisR no further activation can proceed [29]. Expression of the *clc* genes in exponential phase remained unaffected, as expected for being outside the transfer competence regulon. That of the MfsR-controlled *tciR* and efflux pump genes [41] remained constitutive in absence of *bisR* (Fig 6A), indicating they are not influenced by BisR. Transcription from ORF52324 and ORF67800 was also unaffected by the *bisR* deletion, confirming that they are not part of the transfer competence regulon. Taken together, this indicated that the ICE transfer competence promoters are dependent on the early regulators MfsR/TciR, yet only activated through BisR as an intermediate step.

Next we tested whether the suspected ICE*clc* promoters could be directly activated in absence of the ICE in *P. putida* by overexpression of BisDC, which is the previously identified key regulator for controlling and maintaining bistable expression [29]. IPTG induction of plasmid-localised *bisDC* expression in *P. putida* without ICE*clc* but equipped with single copy inserted promoter-fused fluorescent gene reporters resulted for all tested constructs in increased reporter fluorescence compared to an empty plasmid control (Fig 6C), except for the negative control (UR89746) and for the *bisR* promoter. The expression intensities were very

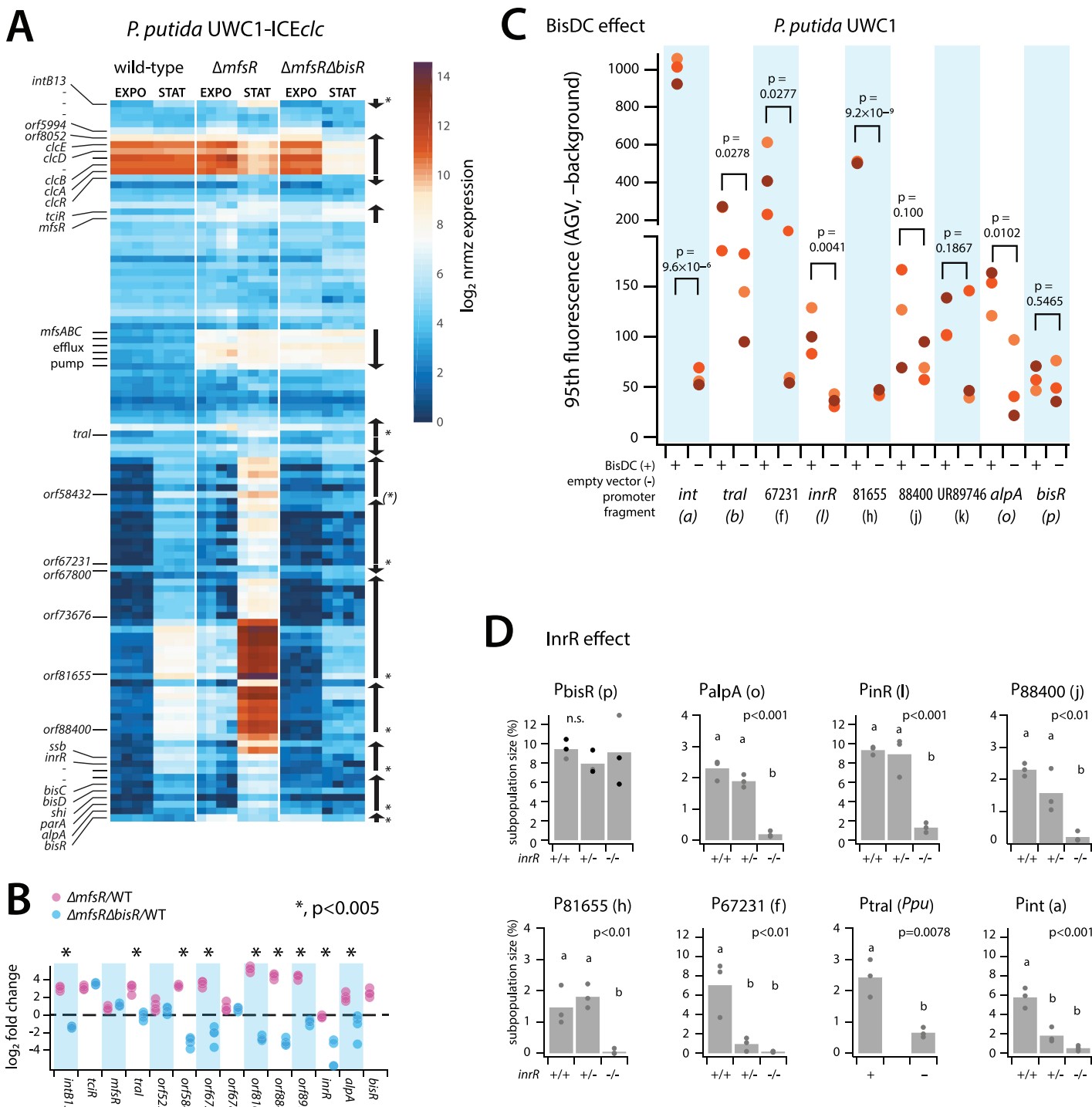

**Fig 6. Dependency of ICE*clc* promoters on ICE global regulators initiating transfer competence. A** Log₂-scaled normalized and per-gene attributed read counts from RNA-seq of *P. putida* carrying wild-type ICE*clc*, Δ*mfsR* or Δ*mfsR*Δ*bisR* deletions, for the ICE*clc* gene region (*intB13* gene on top; *bisR* at bottom). Each colored rectangle corresponds to a single gene (relevant names on the left) and replicate (four replicates per condition and strain; growth on 3CBA, sampled in exponential phase and late stationary phase). Organisation of transcriptional units depicted with arrows on the right (asterisks corresponding to those being part of the transfer competence regulon, role of P₅₈₄₃₂ inconclusive). **B** Calculated log₂-fold changes between mutant and wild-type ICE*clc* for the tested promoter regions and controls. Dots corresponding to individual replicate values. Dotted line represents ratio of 1. Asterisks denote statistically significant differences in *mattest* with bootstrapping (n = 1000). Note that reads could not be mapped to *bisR* in the corresponding deletion mutant. **C** Effect of induction of plasmid-located *bisDC* (+) on reporter fluorescence of *P. putida* without ICE*clc* with indicated single copy inserted promoter- or control (UR89746) fusions. Cells sampled in stationary phase after growth on succinate. Comparisons are the same *P. putida* reporter strains but with empty plasmid (−). Plotted is the 95th percentile scaled fluorescence (minus image

background) of the population of sampled cells ($n$ = 10 technical replicates, each dot from a single biological replicate with independent reporter gene insertion position), to account for the strongly tailed fluorescence distributions. p-values calculated from paired one-sided t-test of strains in presence of BisDC versus the vector-only control ($n$ = 3, replicates per construct). **D** Effect of *inrR* (two, one or no copy) on ICE*clc* in *P. knackmussii* B13 (note: having two identical integrated ICE*clc* copies) or *P. putida* ICE*clc* (P$_{traI}$-construct) on expression of the indicated reporter constructs (in all cases fused to *egfp*, three clones with different single copy insertion positions). Bars show the mean ± one *SD* of the estimated subpopulation sizes (from quantile-quantile plotting) in cultures growing on 3CBA sampled after 24, 48, 72 or 96 h (24 h corresponding to onset of stationary phase). Letters correspond to significance values in one-factorial ANOVA across all samples followed by post-hoc Tukey test.

different among the tested promoters and several resulted in highly skewed or subpopulation-confined expression (hence testing the 95$^{th}$ percentile levels in Figs 6C and S5). These results thus demonstrated that these transfer competence promoters are 'downstream' in the regulatory cascade of *bisR* and the *bisDC* feedback loop (Fig 2B). Finally, we tested whether their expression was dependent on InrR, a previously reported factor contributing to optimal expression of P$_{int}$ [36]. For this, we introduced the single copy promoter fluorescent reporter gene fusions in a *P. knackmussii* wild-type background, and with one (*inrR$^{+/-}$*, i.e., on one of its ICE*clc* copies) or both copies of *inrR* deleted (*inrR$^{-/-}$*, i.e., on both its integrated ICE*clc* copies), and measured subpopulation-dependent fluorescence expression in stationary phase cells of 3CBA-grown cultures (Fig 6D). As before, the *traI*-promoter construct was tested in *P. putida* with one ICE*clc* copy. Apart from P$_{bisR}$, all other promoters were dependent on InrR, with strongly diminished subpopulation sizes in absence of one or both *inrR* gene copies (Fig 6D). Collectively, these results thus indicated that the ICE*clc* transfer competence regulon encompasses a number of 'late' expressed elements (promoters upstream of *inrR*, ORF88400, ORF81655, ORF67231, *traI* and *intB13*). Their expression is dependent on factors produced in the early stages (e.g., TciR, BisR) and from the feedback loop (BisDC, InrR).

As a large fraction of genes in the ICE*clc* core region is highly conserved (e.g., S6 Fig) but still without functional annotation, we produced a number of seamless gene cluster deletions on the ICE*clc* and tested their effect on activation and transfer rates. Deletion of the region from ORF81655-75419 or from ORF88400-84388 had no measurable effect on ICE*clc* transfer from *P. putida* to a gentamicin-resistant isogenic strain, whereas that of ORF74436-68241 abolished transfer completely (Fig 7 and S5 Data). All constructs expressed similar proportions of tc cells in stationary phase (Fig 7), indicating that absence of transfer was not due to abolished induction of transfer competence.

## Discussion

ICE*clc* had been hypothesized to impose a bistable differentiation program that elicits host cells to become competent for ICE transfer [27]. Previous studies visualizing single cell ICE*clc* activation from fused fluorescent protein genes to the integrase promoter P$_{int}$ and the promoter of the integrase regulator gene *inrR* had inferred that this transfer competence would arise in a 3–5% subpopulation of cells carrying the ICE [19,35,36], under stationary phase conditions [37] and most pronounced after growth on 3CBA [35,36]. Since ICE conjugative transfer is assumed to involve a variety of distinct steps (e.g., excision, unwinding to single-stranded DNA and replication, presentation to the conjugation complex [24,27]), we hypothesized that transfer competence development would encompass hierarchically and temporally controlled activation of subsets of ICE-genes in the same individual cells that commit to the complete process. We uncovered here that most of the genes within the conserved core region of ICE*clc* belong to a program that we can now name 'transfer competence regulon' (TCR). The regulon is organized in five, possibly, six 'downstream' transcriptional units that are coordinately transcribed within the same subpopulation of cells (Figs 1 and 2). The only promoter for which

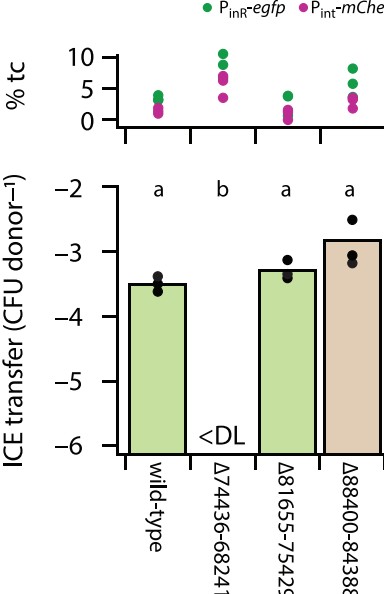

**Fig 7. Effect of ICE*clc* gene region deletions on transfer frequency (bottom) and subpopulation size of P$_{int}$-mCherry/P$_{inR}$-eGFP expressing cells (top, as percentage from qq-plots).** Bars show the calculated mean ICE transfer frequency (as colony forming units of transconjugants per CFU of the donor) for the indicated ICE in *P. putida* (deletions as in Fig 1A), with dots representing individual replicate values. Letters correspond to significance values (p < 0.005) in one-factorial ANOVA followed by post-hoc Tukey test. <DL, below detection limit.

not all approaches gave coherent results, P$_{58432}$, is actually downstream of P$_{67231}$ (Fig 1B). Therefore, even if one would assume it is not functional, these genes would still be part of the TCR. All downstream transcriptional units are dependent on the transcription activator complex BisDC, and on the previously discovered factor InrR [36], which are produced in the feedback loop that is initiated by action of BisR and its 'upstream' cascade of MfsR and TciR (Fig 2) [29]. In addition, the core region has two transcriptional units that are oppositely oriented and do not seem to be part of the TCR (i.e., ORF67800 and ORF52324-53196).

Our conclusion is based on several lines of evidence, notably fluorescent reporter expression in single cells controlled from different individual or pairs of ICE core promoter regions, in ICE*clc* wild-type and mutant backgrounds. RNA-seq further helped to identify global regulatory effects, and differences in read coverage abundances supported attribution of potential promoter regions. The common detected promoter sequence motif corroborates coordinated transcription regulation, although further studies are needed to confirm actual BisDC binding sites. Finally, single cell fluorescent reporter expression was essential to quantify and restrict the subpopulation of stationary phase tc cells, which is a hallmark of the ICE activation program [36]. Quantile-quantile analysis of single cell fluorescence distribution is currently the only statistical tool we have to attribute cells to subpopulations with deviating expression characteristics (the tc cells), but even this cannot be applied to robustly define the exact cut-off between main and subpopulations [42]. Time-lapse analysis of single cell expression further helped to validate tc cell assignment. But this also showed that expression of individual TCR promoters is quite variable, both temporally and in intensity, and marker expression may be too low to be faithfully detected despite a cell in reality going through the transfer competence state. Despite this, and although we may have missed minor promoter details by cloning, we are fairly confident to have uncovered the major transcriptional units of the TCR and their

attribution in the expression hierarchy (Fig 2). Furthermore, although the expression of each of the individual TCR promoters taken at face-value is bimodal within the population of stationary phase cells, pair-wise and temporal expression patterns indicate them to transcribe in the same individual cells. All evidence thus points to the TCR of ICE*clc*, once initiated, imposing a differentiation pathway on a subset of cells only. This pathway leads to cells being able to excise, process and transfer the ICE, and to them arresting cell division upon renewed growth, as was shown elsewhere [18, 19]. All this makes ICE*clc* transfer competence a genuine bistable differentiation pathway, because it fulfills two bistability requirements, namely, operating a set of coherently expressed factors in the same subset of cells and, secondly, leading to a different state from which the cell does not return [43–46].

Among the late genes of the TCR, gene deletion and mating experiments suggested that at least the second half of the multicistronic unit downstream of the P$_{81655}$ promoter is essential for ICE*clc* transfer, whereas genes included in the regions under control of P$_{67231}$ may code for distant type IV secretion system components [39]; and, thus, would be essential for ICE transfer as well. The role of the relaxase gene (*traI*) for ICE*clc* had been shown previously [47] and we confirm here that its expression belongs to the TCR. By contrast, the role of the genes under control of the P$_{88400}$ promoter for ICE transfer is not clear, since their deletion did not result in measurable reduction of transfer frequency from a *P. putida* ICE donor to an isogenic *P. putida* recipient. The genes in this region code for conserved hypothetical proteins leaving little room for speculation as to their potential function in ICE*clc* transfer. However, since many of them are conserved among ICEs of the same family in different hosts both by individual sequence as well as in gene synteny (S6 Fig), it is likely that they have functional importance for some aspect of ICE maintenance, regulation and/or transfer.

One of the intriguing questions in the ICE*clc* TCR pathway is to understand how it can ensure that individual cells, which initiate the pathway, will continue along its path and successfully transfer the ICE? And, secondly: how can it be avoided that parts of the TCR pathway are expressed in non-tc cells that might impact their survival (given that TCR eventually comes with a cost of cell arrest [18,19])? Bacteriophages have solved the problem of orthogonal expression of components, for example, by coding for their specific phage DNA and RNA polymerases [21,22]. As the ICE does not appear to encode such RNA polymerase it needs to accomplish this task differently. Our results suggest that control of the downstream TCR promoters is maintained through two factors (BisDC and InrR, Fig 2), which are both exclusively expressed in tc cells. Time-lapse single cell expression data from individual and pairs of TCR promoters indicated that activation is quite noisy, and is very sensitive to placement of additional promoter copies (i.e., apart from the ICE itself; for example Fig 5A), which is suggestive for promoters regulated by low copy number transcription factors in the cell [48,49]. On the basis of stochastic models, we had previously suggested that the feedback loop (Fig 2A) might function to maintain low but steady levels of BisDC in cells that have initiated the TCR [29], such that its downstream promoters can be activated. Since this is an autoregulatory loop, changes in "free" BisDC levels as a result of promoter binding, would be compensated for by increased production. On the basis of the results shown here (Fig 6), we have to conclude that the proposed feedback loop by BisDC may have an additional component that includes InrR, whose biochemical role is so far not understood. Although overexpression of BisDC is sufficient to activate TCR promoters in absence of the ICE, our current working hypothesis is that InrR under wild-type conditions acts in conjunction with BisDC to provide TCR-promoter specific recognition and recruitment of the host RNA polymerase. This would give the fidelity to the system to follow the transfer competence in the same cells where it initiated.

Another curious discovery here was that expression of the *bisR* promoter is already bimodal, but is visible in 2–3 times the proportion of cells that continue the TCR and express

the downstream promoters. The same stochastic modeling of ICE*clc* regulation had also suggested that the input levels of BisR at the point of initiating the feedback loop (at the *alpA* promoter) are determinant for its output in terms of proportions of cells with active TCR [29]. Furthermore, a synthetic inducible *bisR* construct produced scalable subpopulation sizes of activated cells [29]. The onset of *bisR* and *alpA* expression indeed precedes that of the downstream promoters, but the excess of *bisR*-expressing cells is already reduced at the *alpA* node to the level observed for the downstream TCR genes. What is then the mechanism that, as we discovered here, subdues BisR activation at the *alpA* promoter under wild-type conditions, or rather, seems to lead to abortion of TCR in half of the cells? Without knowing further biochemical details on protein stability and binding constants this is hard to deduce, but possibly also here, the crux lays in the action of InrR as auxilliary protein. We assume, therefore, that it is not only BisDC, but the BisDC/InrR combination that controls the equilibrium of the 2–5% of wild-type cells that develop transfer competence, whereas otherwise the proportion of transfer competent cells would be solely controlled by those expressing BisR [29].

In summary, we uncovered the extent and the hierarchy of the regulon that encompasses transfer competence development of ICE*clc*, showing how the TCR initiates and then restricts to this subpopulation of active cells that are the centerpiece of efficient ICE transfer. ICE*clc* has evolved to a remarkably efficient transfer machine, operating within "the window of opportunity" that it creates in a few individual cells to not disturb its host population (too much) and still transfer highly efficiently [27]. Understanding this process and its adaptation is crucial, given the broad occurrence of ICE in prokaryotic genomes [50], and the particular wide distribution of the ICE*clc* family of elements [28,33], also among important opportunistic pathogens [51,52] with ICE-carried antibiotic resistance genes [30–32]. A further central question to solve is the influence of environmental or physiological cues (such as 3CBA metabolism in case of ICE*clc*) on the proportion of appearing tc cells. Such cues or changes in environmental conditions may unwillingly influence gene transfer rates within microbial communities [53–55], and this may lead to enhanced adaptation of pathogenic isolates to antibiotic resistances carried by the ICE.

## Materials and methods

### Bacterial strains and plasmids

*Escherichia coli* strain DH5α (Gibco Life Technologies, Gaithersburg, Md.) was routinely used for plasmid propagation and cloning experiments. *E. coli* DH5α-λpir was used for the propagation of pBAM plasmids used for the delivery of mini-Tn*5* transposons [56]. The original host harbouring ICE*clc* is *P. knackmussii* B13 [28,57]. *P. putida* UWC1 was used as a further host for (a single copy of) ICE*clc* [47]. Bacterial strains, plasmids and primers used in this study are listed in S1 and S2 Tables, respectively.

### Media and growth conditions

*E. coli* strains were cultivated in Luria Bertani (LB) medium with incubation overnight (O/N) at 37˚C. *P. knackmussii* B13 and *P. putida* were grown at 30˚C in minimal medium (MM) based on the type 21C medium [58] with 5 mM 3CBA or 10 mM succinate as sole carbon and energy source. When required, the following antibiotics were added to the media at the following concentrations: ampicillin 100 μg ml$^{-1}$, kanamycin 50 μg ml$^{-1}$ (*E. coli*) or 25 μg ml$^{-1}$ (*P. knackmussii* and *P. putida*), gentamicin (25 μg ml$^{-1}$), tetracycline 20 μg ml$^{-1}$ (*E. coli*) or 100 μg ml$^{-1}$ (*P. putida*) and chloramphenicol 5 μg ml$^{-1}$. Transcription from the P$_{tac}$ promoter was induced by addition of 0.05 mM isopropyl β-D-1 thiogalactopyranoside (IPTG).

## DNA manipulations

Isolation of chromosomal and plasmid DNA, PCR, restriction enzyme digestion, ligation and electroporation were performed as described by standard procedures [59], and as previously described [29]. Electrotransformation of *P. knackmussii* and *P. putida* was performed using the procedures described by Miyazaki et al. [47]. Seamless chromosomal deletions on ICE*clc* were produced using I-SceI-induced chromosomal breakage and double recombination, as described [29, 60].

## Reporter gene fusions

Appropriate DNA fragments containing the putative ICE*clc* promoter regions ([39]; S2 Table and S1 Fig) were amplified by PCR and cloned in front of a promoterless *egfp* gene on the mini-Tn*5* delivery vector pBAM [56]. The resulting pBAM-eGFP promoter reporter fusions were then introduced in single copy onto the chromosome of *P. knackmussii* B13 or *P. putida* UWC1 (ICE*clc*) using electrotransformation. Transformants were selected on Km selective medium and verified by PCR for appropriate integration. For each reporter fusion, at least three independent clones were selected and purified, with which microscopy analysis of eGFP expression was carried out.

For promoter pair studies, a mini-Tn*5* containing either of the different promoter-*egfp* constructs was inserted in *P. knackmussii* B13 or *P. putida* UWC1 (ICE*clc*) already containing a $P_{int}$-*mcherry* reporter fusion (S1 Table). The resulting transformants were selected on Km and Tc selective medium and verified by specific PCR; and three independent clones were stored.

## Epifluorescence microscopy

For the detection of eGFP and mCherry expression in single cells, *P. knackmussii* B13 or *P. putida* strains were cultured for 16 h at 30˚C in LB medium. Aliquots of 100 μl of this culture were then diluted in 20 ml MM plus 5 mM 3CBA and corresponding antibiotics, and incubated at 30˚C. After 24 h, 48 h, 72 h and 96 h (cultures typically reach the onset of stationary phase between 24–48 h), a culture aliquot of 400 μl was drawn. The cells in the sample were harvested by centrifugation at $7000 \times g$ for 2 min, after which the cell pellet was carefully resuspended in 50 μl of fresh MM without carbon source added. An aliquot of 4 μl suspension was then spread onto a regular microscopy slide, precoated with 0.7 ml of a 1% agarose in MM solution. Slides were covered with a 50 mm × 15 mm cover slip, moved to the dark room, after which cells were imaged (typically within 15–30 min after application; aiming to have between 5–10 images with $n = 1000$ cells in total per clone and time point).

Images were taken under phase-contrast (10 ms), eGFP fluorescence (500 ms), and mCherry (500 ms) using a Zeiss Axioplan II imaging microscope with a 100× Plan Apochromat oil objective lens (Carl Zeiss, Jena Germany) and equipped with a SOLA SE light engine (Lumencor, USA). A SPOT Xplorer slow-can cooled charge coupled device CCD camera system (1.4 Mpixel; Diagnostic Instruments, Sterling Heights, Mich.) fixed on the microscope was used to capture the images. Cells on images were automatically segmented using Super-Segger [61] as previously described [12], calculating average per cell fluorescence intensities in the eGFP and/or mCherry channels. Subpopulations of tc cells were inferred and quantified using quantile-quantile-plotting, as described by Reinhard [42], which is rather insensitive to differences in fluorescence intensities. In short, this entails defining the fluorescence distribution of the main population between first and third quartiles, inferring the linear regression line through the main population on the qq-plot, adding an upper 95% confidence interval on the slope, and quantifying the subpopulation from cell fluorescence values above this slope-confidence interval [42]. Note that the qq-plot method becomes rather inaccurate when the

'sub'-population is very large [42] (although we still used it for comparison in Table 1 data). The procedure was implemented in a custom-made MATLAB script that additionally calculated the mean fluorescence background of the image, the mean fluorescence of the main population and of the subpopulation (of tc cells) (vs 2016a, MathWorks). To compare across ICE promoters, fluorescence values (F) were normalized by subtracting the image background (I) and divided by the same (i.e., (F–I)/I), and then averaged across three independent clones (data reported in Fig 3D). Significance of increased expression in the subpopulation of tc cells was tested on the three paired normalized values (one-sided paired t-test, $H_1$ assuming that tc subpopulation expresses higher than main population). Reporter fluorescence values in paired promoter combinations were additionally normalized to the maximum value in each fluorescence channel (i.e., (F-I)/(Fmax-I)*100; scatter plots of Fig 4).

## Time-lapse experiments

For time-lapse experiments, *P. knackmussii* B13 and *P. putida* strains were precultured in LB, and then grown in MM with 3CBA and appropriate antibiotics, as described above. After 96 h incubation at 30°C, enough for cells to reach stationary phase and produce tc cells, the culture was diluted 100-fold in MM without carbon substrate added and transferred to microscope growth medium surfaces ("gel patches"). Four gel patches (volume each 0.13 ml, 1 mm thick and 6 mm ø) were cast in a microscope POC chamber, as described previously [42]. Gels contained 1% *w/v* agarose in MM with 0.1 mM 3CBA. Three patches were seeded with 6 μl of the 100-fold diluted cell suspension, left to dry at ambient air for 3–5 min in a laminar flow hood, then turned upside down and placed on round cover slip (42 mm ø, 0.17 mm thickness). A silicon spacer ring (1 mm thickness) was added and a second circular cover slip was put on top, after which the whole system was mounted in a rigid metal cast POC chamber and fixed with a metal ring [62]. The POC chamber was incubated at 21°C and images were taken (PhC, eGFP and mCherry) with a Plan Apo λ 100× 1.45 NA Oil objective during 48 h with intervals of 30 min at eight random positions using a Nikon Eclipse Ti-E Inverted Microscope, equipped with a Perfect Focus System (PFS) and pE-100 CoolLED illumination. In between imaging, the microscope lense was "parked" at the unseeded patch, in order to avoid illumination/heat damage to the cells. An in-house program written in Micro-Manager 1.4 was used to pilot the time-lapse experiments and record image series. Images were subsequently processed as described above, to automatically segment and position the cells across time-series, and to extract eGFP and mCherry per cell average fluorescence values. Cell identities given during segmentation were used to align corresponding eGFP and mCherry fluorescence profiles. Cell traces (*n* = 10,000–30,000 per promoter pair) were plotted to define the fluorescence drift in the main population of cells, and define a stationary phase subpopulation threshold by qq-plotting as described above. This threshold was then imposed on the complete data set to remove tc cells carried over from the preceding stationary phase (time span 0–10 h, in exponential phase). Cells were connected to their mothers to quantify cell population growth on the patch (i.e., blue dashed lines in Fig 5A). A moving qq-plot threshold was then calculated on each time point and for each fluorescence individually, to determine the increase of the assigned tc cell population size over time and visualize temporal subpopulation promoter expression differences (i.e., lines in Fig 5B). These two subsets of cell IDs were combined (since tc cell determination on one fluorescence not necessarily overlaps with the other channel), and further filtered to encompass those with at least ten time points. Finally, on this thresholded subset of cell traces, the slopes of fluorescence intensity change (for each channel individually) at each time point was calculated as the linear regression during at least five consecutive time points. The distribution of these values was then used to define the threshold

between spurious and 'real' trace fluorescence increase (using traces as in Fig 5A), and the minimum time point for each trace at which slopes with an $r^2 > 0.9800$ surpassed the threshold was retained. Data sets with less than 100 cells were further inspected interactively on individual traces in comparison to the moving qq-plot thresholds and confirmed if slopes contained more than one time point. The resulting list of paired starts was then plotted pair-wise (i.e., data in Figs 5C and S4). Absence of detected start in one of the fluorescence markers was arbitrarily set at a value of 1 to allow plotting. Note that we did not take fluorescent protein maturation time into account for comparison of expression onsets between individual ICE promoters.

## ICE*clc* transfer

ICE*clc* transfer assays were carried out as described elsewhere [29]. In brief, *P. putida* ICE*clc* donors were cultured for 96 h in MM with 5 mM 3CBA (plus appropriate antibiotics to select for genetic constructions) to induce tc cell formation, whereas recipient cultures (*P. putida* UWCGC, gentamicin-resistant derivative of UWC1) were grown for 24 h in MM with 10 mM succinate and gentamicin. Recipient and donor cultures were mixed in a 1:2 volumetric ratio, respectively, in a total volume of 1 ml. Cells were harvested by centrifugation at room temperature for 1 min at $5000 \times g$, washed in 1 ml of MM without carbon substrate, centrifuged again and finally resuspended in 20 µl of MM. This mixture was deposited on top of a 0.2–µm cellulose acetate filter (Sartorius) placed on MM-agar containing 0.5 mM 3CBA, and incubated at 30°C for 48 h. Cells were then recovered from the filter by vortexing in 1 ml of MM, serially diluted in MM, and plated on selective plates. The same culture volumes of either donor or recipient alone were prepared and incubated similarly, to correct for the frequency of spontaneous background growth. Exconjugants were selected on MM agar plates with Gm and 3CBA (from transfer of ICE*clc*); donors on MM with 3CBA and Km, and recipients on MM with Gm and 10 mM succinate. Transfer frequencies are reported as the mean of the exconjugant colony forming units on MM-Gm-3CBA compared to that of the donor in the same assay (on MM-Km-3CBA).

## RNA-seq

Total sequencing of reverse-transcribed rRNA-depleted mRNAs (RNA-seq) was conducted on exponentially growing or 'restimulated' stationary phase cultures of *P. putida* UWC1 carrying wild-type ICE*clc* (strain 2737), ICE*clcΔmfsR* (strain 4322), or ICE*clcΔmfsRΔbisR* (strain 5553). Cultures were grown in fourfold replicates in MM with 5 mM 3CBA as described previously [63] and harvested in exponential phase at a culture turbidity of 0.6 (at 600 nm). Four other replicate cultures were incubated for 96 h on 5 mM 3CBA (late stationary phase, to induce TCR), and then stimulated for 4.5 h by addition of 5 mM 3CBA (final concentration) to induce ICE excision and transfer. Cells were harvested by centrifugation as described, and total RNA was purified by hot phenol, DNAseI digestion, MiniElute cleanup, and depleted from rRNAs using the Ribo-Zero rRNA removal kit (EpiCentre) [63]. cDNA libraries were generated using a strand-specific ScriptSeq Complete Kit Bacteria protocol (Epicentre), indexed and sequenced on an Illumina HiSeq 2000 platform at the Lausanne Genomics Facilities with 101-nt single-end reads. Reads were cleaned and trimmed using TRIMMOMATIC [64], then mapped, sorted and indexed using Bowtie2 [65] and Samtools [66] under default settings, using the *P. putida* strain KT2440 chromosome (refseq NC_002947) and ICE*clc* (Genbank accession AJ617740.2) reconstructed genome sequence as reference. Mapped reads were counted with HTseq (version 0.11.2) [67]. Read counts were normalized using the PseudoReference Sample transformation [68]. In short, for each gene the geometric mean across all samples was

calculated and used as the pseudoreference sample. Then for each sample, the read count of every gene was divided by its corresponding value in the pseudoreference. The median value of all those ratios of a given sample was used as the normalization factor for that sample. Data were $\log_2(x+1)$ transformed in order to deal with zero values, before further clustering using *clustergram* as implemented in MATLAB (v. 2020a). For coverage plots, raw HTseq counts per position from a single replicate condition were read into MATLAB and plotted in a select window of 1000 bp covering the gene of interest. Plots were overlaid for two conditions in Adobe Illustrator (v. 2020).

## Sequence motif search

Promoter motifs were searched by MEME (Multiple Em for Motif Elicitation) [69], by using as input the identified bimodal TCR promoter regions (300 bp input fragment). The identified 27-bp motif was then used in FIMO (Find Individual Motif Occurrence) and MAST (Motif Alignment and Search Tool, all within the MEME package) to screen ICE*clc* for further occurrences (no other significant hit). The *traI* promoter region was aligned manually to S3 Fig.

## Statistical methods

Mean background-normalized fluorescence expression values for different single-copy promoter fusions were compared between main and subpopulations of tc cells ($n = 3$ replicates with different clones, paired one-sided t-test). Effects of BisDC induction on ICE promoter expression in absence of ICE*clc* in *P. putida* was tested on $n = 3$ independent replicates with different mini-Tn inserted promoter fusions, grown to stationary phase (48–96 h) on MM with 5 mM 3CBA. Because of strongly skewed fluorescence distributions we took here the 95th percentile value to compare between strains carrying *bisDC* or with empty plasmid (paired one-sided t-test). Effects of InrR on the subpopulation sizes of cells expressing fluorescence from single-copy ICE promoters was tested on $n = 3$ independent replicates with different mini-Tn inserted promoter fusions, grown to stationary phase on MM with 5 mM 3CBA and sampled at 24, 48 and 72 h. Triplicate estimates of subpopulation sizes (by qq-plotting) across the three strains were then compared in one-factorial ANOVA, implemented as the Bartlett test in *R*, followed by *aov* and Tukey multiple comparisons of means at 95% family-wise confidence level. The same procedure was followed to compare transfer rates among ICE-deletion variants.

## Supporting information

**S1 Table. Strain specifications.**
(DOCX)

**S2 Table. Primers used.**
(DOCX)

**S1 Fig. Cloned and tested upstream/promoter fragments of ICE core genes.** A General overview of the ICE core region. B Individual selected promoter/upstream fragments and their sizes (small cap letters corresponding to fragment indications in the main text).
(PDF)

**S2 Fig. Read coverage of ICE*clc* transcription in *P. putida*.** A) Plots show regions tested for promoter activity with read coverage per basepair position from RNA-seq (for a single representative replicate) at the indicated conditions (3CBA, exponential phase in black; stationary phase in green), plotted for the relevant *P. putida* genome region with the integrated ICE*clc* on

the x-axis (in Mbp). B) Read coverage of ICE*clc* transcripts in *P. putida* ICE*clc* in stationary phase conditions after growth with 3CBA (green) or succinate (brown) as carbon substrate. Blue lettered bars point to cloned fragments tested for promoter activity at single cell level. Dotted black arrows point to subpopulation-dependent tc cell promoters; straight lines when expressed in all cells. Open directional bars ($<$ or $>$) correspond to relevant coding regions on ICE*clc*. P*circ*, outward facing constitutive promoter.
(PDF)

**S3 Fig. Common sequence motif in the identified transfer competence promoters of ICEclc.** Motif identified by MEME. Sequence of $P_{traI}$ added manually to align. No other similar motif was found on ICE*clc*. Distance indicated to the start codon of the downstream gene. Distance to the mapped transcription start site in the $P_{inR}$-promoter: 152 bp.
(PDF)

**S4 Fig. Timing of onset of transfer competence promoter expression in *P. knackmussii* (A-G) and *P. putida* ICE*clc* (H).** Single copy inserted promoter-reporter fusion pairs as indicated on top of each panel, with color legend. Left panels show global increase of the inferred tc cell population from the respective marker. The right panels show paired automatically detected onsets of fluorescence increase in individual tc cells. Values of 1 are artificially attributed when no slope was detected for the respective cell and fluorescence reporter. See further, explanation to Fig 5 in main text. Note that panel D global increase could not be quantified because of drifting image focus between time 19 and 20 h.
(PDF)

**S5 Fig. Cell fluorescence distribution from indicated single copy promoter-reporter fusions in *P. putida* without ICE*clc*, but induced or not for production of the BisDC activator complex (pMEbisDC).** Cells sampled in stationary phase after growth on succinate. Comparisons are the same *P. putida* reporter strains but with empty plasmid (pME6032). Cell fluorescence distributions are plotted as their expected versus observed quantile; each plot showing a single biological replicate with independent reporter gene insertion position, grouped from n = 10 images per sample. Each dot corresponds to a single segmented cell observation. Note the strongly tailed distributions for some constructs.
(PDF)

**S6 Fig. ICE*clc* gene and gene synteny conservation to putative ICE in genomes of other Gamma- and Betaproteobacteria.** Regions are aligned to the gene cluster containing *inrR* and *ssb* (ochre), and then emphasize the conserved regions with unknown functions orf88400—orf81655. Ortholog genes are colored similarly. Arrows indicate the corresponding open reading frame length and orientation. Numbers below represent the percent nucleotide identity to ICE*clc*. White open arrows point to open reading frames not generally conserved with ICE*clc*.
(PDF)

**S1 Data. Source data for Fig 3.** Quantification of tc cell subpopulations.
(XLSX)

**S2 Data. Source data for Fig 4.** Quantified colocalized paired bimodal promoter fluorescence expression.
(XLSX)

**S3 Data. Source data for Fig 5.** Quantified time-lapse fluorescence expression of paired promoters from the ICE*clc* transfer competence regulon.
(ZIP)

**S4 Data. Source data for Fig 6.** Dependency of ICE*clc* promoters on ICE global regulators, RNA-seq and expression data.
(XLSX)

**S5 Data. Source data for Fig 7.** ICE*clc* transfer data.
(XLSX)

## Acknowledgments

We thank Noémie Matthey for her help in initial parts of this project.

## Author Contributions

**Conceptualization:** Sandra Sulser, Andrea Vucicevic, Roxane Moritz, Jan Roelof van der Meer.

**Data curation:** Sandra Sulser, Andrea Vucicevic, Veronica Bellini, Roxane Moritz, Nicolas Carraro, Jan Roelof van der Meer.

**Formal analysis:** Veronica Bellini, François Delavat, Vladimir Sentchilo, Nicolas Carraro, Jan Roelof van der Meer.

**Funding acquisition:** Jan Roelof van der Meer.

**Investigation:** Sandra Sulser, Andrea Vucicevic, Veronica Bellini, Roxane Moritz, François Delavat, Vladimir Sentchilo, Nicolas Carraro, Jan Roelof van der Meer.

**Methodology:** Sandra Sulser, Andrea Vucicevic, Roxane Moritz, François Delavat, Vladimir Sentchilo, Nicolas Carraro.

**Project administration:** Jan Roelof van der Meer.

**Software:** Roxane Moritz, Jan Roelof van der Meer.

**Supervision:** Vladimir Sentchilo, Nicolas Carraro, Jan Roelof van der Meer.

**Validation:** Nicolas Carraro, Jan Roelof van der Meer.

**Visualization:** Roxane Moritz, Nicolas Carraro, Jan Roelof van der Meer.

**Writing – original draft:** Nicolas Carraro, Jan Roelof van der Meer.

**Writing – review & editing:** Sandra Sulser, Andrea Vucicevic, Veronica Bellini, Roxane Moritz, François Delavat, Vladimir Sentchilo, Nicolas Carraro, Jan Roelof van der Meer.

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
