## [Decision Letter · Decision Letter 0]

1 Feb 2022

Dear Dr van der Meer,

Thank you very much for submitting your Research Article entitled 'A bistable orthogonal prokaryotic differentiation system underlying development of conjugative transfer competence' to PLOS Genetics.

The manuscript was fully evaluated at the editorial level and by independent peer reviewers. The reviewers appreciated the attention to an important problem, but raised some substantial concerns about the current manuscript.

Reviewers have identified problems concerning novelty, terminology, data presentation and interpretation. Some of these problems can be resolved by significant rewriting. However, particularly important problem is a difficulty to determine what is new information from what was previously known. As stated by Reviewer #2, many of presented data are from the new experiments that confirm previous findings obtained by somewhat different experimental approaches. The new information is essentially that at least six promoters in ICEclc are expressed predominantly in the subpopulation of cells that are transfer competent in stationary phase, and that they require the same regulators. However, much of this was documented by your lab for the promoter for int (Pint). In order to show that the six or so promoters are part of the same regulon, more molecular analysis of direct regulators is required.

Based on the reviews, we will not be able to accept this version of the manuscript, but we would be willing to review a much-revised version. We cannot, of course, promise publication at that time.

If you decide to revise the manuscript for further consideration at PLOS Genetics, please aim to resubmit within the next 60 days, unless it will take extra time to address the concerns of the reviewers, in which case we would appreciate an expected resubmission date by email to plosgenetics@plos.org.

[LINK]

We are sorry that we cannot be more positive about your manuscript at this stage. Please do not hesitate to contact us if you have any concerns or questions.

Yours sincerely,

Ivan Matic

Associate Editor

PLOS Genetics

Lotte Søgaard-Andersen

Section Editor: Prokaryotic Genetics

PLOS Genetics

Reviewer's Responses to Questions

**Comments to the Authors:**

Reviewer #1: The manuscript by Sulser et al. describes how six gene clusters of an integrative conjugative element (ICE) are activated in a small subset of the cells (<10%). By using a dual labelling strategy (GFP/mCherry) they show that in most of these clusters, the subset of cells that activate transcription correspond to cells transcribing the integrase gene. These results suggests that activation of the ICE promoters is correlated with integrase expression, and ICE conjugation. Previous results have shown a bimodal activation pattern of ICE conjugation: upon entrance in stationary phase, only a small fraction of the cells actively transcribe certain key transcriptional activators (BisR, InR), which result in transfer competence. By looking at the expression levels of the ICE promoters in different mutants, the authors show that these genes constitute the final part of the activation program leading to ICE conjugation. Overall, the paper is well written and experimentally sound. Results presented here contribute to completing the picture of the ICE activation program, thus these findings are of broad interest to understanding the dynamics and physiology of these mobile genetic elements. My only concerns refer to the use of certain terminology and data presentation issues, which I think should be addressed for a better, more rigorous presentation of results.

Minor Comments

1.- My main concern is the rather lax usage of two terms: bistable and orthogonal. From the title to the discussion, these two buzzwords are used frequently in the paper, yet I am afraid their employment is not entirely justified by results presented here. A bimodal distribution is not the equivalent of a bistable system. The latter tends to produce (but not always) the former, and bimodality may be achieved by different mechanisms that do not necessarily imply bistability. I have no objection to mention the possibility of this being an example of a bistable system in the discussion -it may well be one- but results presented do not justify the claim of the title. I might be missing something, but in the references cited by the authors there is neither much evidence of the usual hallmarks of bistability (positive feedback, hysteresis, etc…) nor any stability analysis proper, which could justify the claim. The strongest evidence of a possible bistable mechanism is the cell arrest phenotype demonstrated in Ref18. However as shown in figure 1C of the manuscript, the promoter for parA/Shi, responsible for cell arrest, is not bimodal. I am also a puzzled by the usage of ref 42 as a justification for bistable behavior. This paper from Leibler mostly deals with stochastic switching as an alternative to deterministic switching (for example by a bistable activator). If anything, it is a demonstration of how other mechanisms may produce bimodal distributions. Confusion between bistability and bimodality is widespread in the literature, thus this may sound like a finicky request. However, the differences are important for people working in the dynamics of regulatory systems. I have also the feeling that some authors tend to use bistable because they perceive this as a more “sophisticated” or valuable mechanism. It is not: many bistable systems never achieve bimodaility. I would suggest the authors to adhere to bimodality, which I think is the most correct term describing their results. Indeed, one possible mechanistic explanation for their results could lie in two critical regulators being stochastically activated and working in a concerted AND fashion (BisR + InrR? or maybe RpoS?). This could result in a bimodal distribution without the need of a positive feedback loop or any other bistable behavior.

The second term that I think should be used with caution is "orthogonal". In particular, I would suggest the authors to clearly state which two aspects/things are being considered orthogonal when the term is used. Otherwise the wording results confusing, as for example in the abstract. There, the authors speak of “six gene clusters coordinately and orthogonally expressed in the same cell subpopulation”. Yet this is patently absurd: if they are coordinated they cannot be orthogonal to each other! My guess is that the authors were trying to indicate that these genes are orthogonal to the host regulatory network. This may well be the case, but I find it difficult to prove, given that RpoS, as indicated by the authors, is a key player in the circuit. Orthogonal just merely indicates that two variables are independent of each other, but this most often gives little to no insight. Moreover, in this case results actually may hint at the opposite. As shown in Figure 6 the activation dynamics is quite different in cells growing in exponential phase, compared to stationary cultures. This could be caused just by the effects of protein dilution/concentration in growing/arrested cells (thus a circuit truly orthogonal). However this could be also caused by RpoS activation, or any other regulator of the stationary phase. From the data presented here, I don´t think we can favor any of the two options.

2.- I really liked the quartile plots in figure 2, which clearly show the activation of a fraction of the population beyond the statistical expectation, as judged from the rest of the distribution. However I found panel 4d a bit clumsy. My guess is that the authors were trying to show the level of induction in the “active” population, compared to the inactive population. Thus they show the averages in both sets. The problem is that , since the active population is defined as the set of cells in the upper quartile of the distribution, the reasoning becomes circular. The average of the upper quartile of any distribution is always higher than the average of the lower quartiles, by definition. To show the information that the authors want to convey here , one alternative could be to show the distribution , deconvolve, and calculate the median/mode of each subpopulation. Alternatively, and probably much easier, the authors could just show the distribution on an x scale, showing the quartiles, so the reader may have an intuition of the magnitude of the increase in the "active" population.

3.- I am also a bit puzzled by figure 4. The distribution of Pint levels (measured from mCherry) should be more or less the same in all panels, right? However the scale on the y axis in the figure suggests otherwise (Pint goes from 90 to 160 when measured with P88400, but from 400 to 1400 when measured with P58432). Are the exposure times / lamp intensities changing between experiments? Does this affect to the definition of Tc competent cell? (my guess is not, because it is quartile-based, but it should be stated in the text).

4.- I also found figure 5 a bit confusing. Panel D, for example, gives little information , thus I think it may be better suited for supplementary material. Panel A, as it is now, is really hard to decipher because the high number of cell traces included. I am not entirely sure how this could be improved, but I would recommend the authors to reduce the burden of this panel. This could be achieved by only including cell trajectories that do show activation ( we already know that up to 90% are going to never activate). From the subset of cells that activate, I would pick a representative sample (maybe 5-10) and plot them in different colors. This way we may be able to check whether the cell that quickly fires in the Pint-mCherry promoter increases simultaneously, earlier or later the GFP trace. Regarding the temporality of activation, I would also recommend caution in the interpretation: mCherry matures in 15-30 minutes in E.coli , while the GFP does in 3-7 minutes. These differences may blur a co-activation temporal patterns. Also, some terms deserve better explanation. For example how is “start time” defined. Also, in panel D the GFP/mCherry counts are not well explained, do they correspond to the “start time”, the maximum level?

Reviewer #2: This manuscript builds on previous work characterizing regulation of ICEclc and formation of a transfer competent (tc) subpopulation of cells of Pseudomonas species. It presents a large amount of work and analyses, including single cell analyses of expression of several promoters in ICEclc and RNA-seq analysis of cells in stationary phase, a subset of which are expressing ICEclc.

The topic is interesting and important. However, I found it quite difficult to determine what is new information from what was previously known. My sense is that many of the new experiments confirm previous findings obtained by somewhat different experimental approaches.

From what I can tell, the novel findings here appear to be that nine promoters in ICEclc are all expressed in transfer competent (tc) cells and at least six of these appear to have coordinate regulation in stationary phase. Previously identified transcripts were tested and found to have bimodal expression in stationary phase. The gene for the regulator BisR was found to be expressed in twice as many cells in the population than ultimately commit to being transfer competent. Additionally, expression from all six promoters was dependent on the previously identified regulators InrR and BisDC. Expression from each promoter was measured in combination with expression from Pint in the same cells, demonstrating that they all turn on in the roughly the same subpopulation of cells in stationary phase, and roughly the same time. They infer that these promoters are part of the same transfer competent regulon.

I had to read this manuscript several times to understand and extract the main points and determine which were the key experiments. There were many experiments presented, and at times they seemed disjointed. I think some serious editing could make this clearer for readers not intimately familiar with ICEclc.

1. Regarding the apparently coordinate temporal control of the six promoters that are expressed in tc cells; It is clear that they are expressed in the same cell type, and not so much in others. I think we knew from first principles that if they are required for conjugation, then they must be expressed in the same cells. What seems new is that expression is largely limited to the tc cells with little expression in the non-tc population.

2. What is the resolution for determining coordination? If a gene in one of the transcription units was an activator of one of the other transcription units, is there the temporal resolution to see what could be an offset in transcription of 5-15 minutes (or less)?

3. It is not clear that these six genes are all part of the same regulon, at least directly. They all require BisDC and InrR, but there could be a hierarchy (see above).

4. Regarding InrR and BisDC. Clearly both are needed. Could the only role of InrR be to activate BisDC? It seems that overproduction of BisDC activates gene expression in the absence of ICEclc (no inrR), perhaps bypassing the need for InrR?

5. Are there similar DNA sequences in each promoter region that could represent potential binding sites for one or more regulators?

6. Fig4; Is there a mistake in 4d with PinR/Pint with PinR on the y-axis? all the other panels have Pint on the y-axis. Also, the reporter is indicated as Pint-eGFP (x axis) whereas all the others use Pint-mCHE (on the y-axis).

7. also fig4; It seems that the fluorescence from Pint-mCHE is different between several of the panels: ~150-200 in a; ~50-100 in c; ~400 in f, etc. What is the cause of these differences? Should they not all be similar?

8. p10, line 14; regarding bisR expression in ~2x more cells than expression from other promoters; this could indicate additional regulators as suggested. It could also indicate threshold effects for BisR, as implied in the discussion.

9. p7; regarding the RNA-seq data; My sense is that these data largely confirm previously published work from Northern blots and microarrays. Please indicate what is added here that was not previously published. Perhaps a better understanding of the location of the 5'-ends of the mRNA? These are consistent with the promoter activities measured with fusions. Also, please be careful not to imply that the RNA-seq data is measuring RNA synthesis, or even transcription start sites. The data identify the approximate location of 5'-ends and RNA abundance (not synthesis).

10. The experiments seem to switch back and forth between P. knackmussii and P. putida, and at least one promoter appears to be active in P. putida and not P. knackmussii. Is there a reason for switching between the two organisms for different experiments?

11. I find the use of 'orthogonal' quite confusing and unnecessary. In some places, I'm not even sure what it means. For example; abstract, line 12; 'coordinately and orthogonally expressed'; orthogonal to what? not each other as they are 'coordinate'. There is differential expression relative to other ICEclc genes/promoters, and perhaps to many chromosomal genes, but perhaps not all.

12. Similarly, intro, line 11; 'maintain orthogonal components'; I think this is simply that MGEs have their own program(s) of gene expression. Different regulons within any bacterium have their own program of gene expression, and for a given regulon, there is some coordination and this is different from expression of other regulons. NO need to use 'orthogonal', which in my opinion complicates a very simple concept.

13. Abstract, line 6; change 'stands' to 'is a'; similarly, p4, line 17; 'stands model' should be 'stands as a model' or, more simply 'is a model'

14. Table 1. regarding the proportion of stationary phase cells expressing reporters; clearly, MfsR is a repressor; but some promoters still only expressed in subpopulation in ∆mfsR cells, and in very different subpopulations. What does this mean?

15. Table 1, footnote e; 'data reproduced from ref 58'; are the data taken from ref58 and shown here for comparison? or was the same experiment redone? I suspect that the data were taken from ref 58 (I did not check).

16. It would be very helpful to put Fig. 8 early, perhaps as Fig2, and use it to highlight what was known, and preview what is to come.

Reviewer #3: This MS describes the transcriptional program of ICE activation for DNA transfer. It is a complex study building on previous work suggesting that ICE activation was limited to a small proportion of the bacterial population by a series of bistable switches. The authors use an impressive array of experimental approaches to try and identify the transcriptional cascade necessary for commitment to DNA transfer. Their most compelling data uses a set of fluorescent transcriptional reporters that are expressed from ICE promoters specifically activated in stationary phase. They further identify several promoters that are coordinately expressed in the same sub-population of cells suggesting a defined synchronized activation pathway for key conjugation genes.

The hypothesis makes sense – if cells are to conjugate then activation needs to be coordinated through a series of independent, parallel steps that ensure the cell is fully committed for trasnfer. The data in general are supportive of the model, but the individual experiments are not convincing and, thus, raise doubt about the conclusions drawn and the overall proposed mechanism of activation.

Below I have focused on the figures as discussion points.

Figure 2 sets the stage for the reporter system by identifying 7 promoters that appear to be bimodal. The scales in 2b vary dramatically from 100-1000, especially for the P.putida data set and 50240. This particular gene reporter is an outlier in many subsequent experiments, and it is not clear to this reviewer if these data can be properly compared with the other promoters in a different strain background. Why are the fluorescent levels so different? They do not correlate with promoter strength.

Fig 3 presents RNA-seq data but the interpretation of these is not clear in the text. The data are poorly presented. Transcript reads are not assigned to a strand, so it is impossible to determine directionality. The read coverage axes are very different and, in many cases, not adjusted to the reads of the relevant gene. E.g., P50240 scale at 0-800 is set to a gene in the opposite direction, when 50240 is expressed at very low levels, similar to int, while the Pint scale is appropriately set to 100 reads. Many of the views are not optimized for read depth. The 5’ of RNA-seq should identify the transcriptional start site. In many cases the “promoter arrow” does not coincide with the 5’ end of reads. Why? Eg., PinR, P81655, P50240, PalpA. Is the precise TSS really known or are these best guesses? The precise identification of each TSS would also provide confidence that the correct a-r promoter segments have been cloned (Fig. 1b, P7 25- and P13 21-23).

RNA seq cannot distinguish between new starts and readthrough making assigning promoter activity very difficult (e.g., 58432, see P7 line 20-21). Similarly, readthrough in RNA seq does not allow one to rule out promoter activity in UR62755,66202 etc. (P8 1-3). As these promoters are activated only in 2-5% of cells it is impossible to interpret reads in tc cells vs the overall population. The authors mention this in passing, but fail to use this knowledge in their data interpretation. The RNA seq data show that essentially all of these regions are transcribed preferentially during stationary phase, regardless of whether the region contains a putative tc promoter (P7, 14-17, p8, 17-19).

Fig 4. The scales for Fig4 are highly variable, and yet all are compared to Pint – the internal control. Mcherry Int-positive cells increase from a cut-off of 105 (c) to 180 (a) and 500 (f). Why?

Label on (d) is switched compared with axes.

The data for 4a-e are compelling, but f-g are less clear and suggest f-g are not coordinately expressed with Pint. Yet P9;10-11 states that only bisR is not coordinately expressed. Authors suggest that 58432 is not clear because of it having a weaker promoter. But its promoter is as strong as 81655 and 88440 (Fig 2d), so this argument appears flawed. 50240 is an outlier again as it is only expressed in Pput. Despite having made the argument for inclusion of 50240 and 58432 as part of the tc regulon (p9 top paragraph), the authors exclude these two promoters from further analysis because of their poor correlation (P9 bottom paragraph). This is appropriate given their poorer correlations, but the authors still include these two genes in their discussion and model regulon (Fig 8).

Fig 5 and S3. Data for 67231 are not included in Fig 5 or fig S3 despite it clearly belonging to the tc regulon. These are complex figures that need better explanation and presentation. It is clear that bisR is different from the others. What is not clear, especially in Fig 5a, is the co-expression of fluorescence in both cells because the figs are too noisy. S3 does this a bit better as the traces are labelled. Perhaps a more convincing approach would be to provide a few graphs with no background and showing just a few cells with onset for both reporters?

5d/e need more explanation as I came to a very different conclusion to the reviewers on P10;3-15, when they suggest that the timing onset for all was well correlated except for BisR. What seems clear to me is that Pint and P81655 exhibit early expression in both 5a and S3a with expression onset in the 5-10 hr window. By contrast for the other three constructs, Fig S3 indicates their first fluorescent response is in the 1000-1500 min window (16 hrs). This is especially clear for S3c/d, in which there are no signals in the first 1000 mins. This observation is further supported in 5de, when comparing 81655 and int to other promoters; their first brown squares (d; left graph) co-express at 10 hrs vs 20 hrs for all others. The histograms in (e) follow the same pattern. The interpretation of this figure is key to the overall model (fig 8, P12,9-13 and discussion), which posits that all of these promoters are coordinately induced at the final stage of the regulon. My interpretation of the data suggests that Pint and 81655 are induced earlier.

Additional issues with these two figures

5c, could the authors confirm the PalpA line of correlation is correct?

S3 needs correcting. Time is in mins not hours. The scales vary widely, even for mcherry. Why?

Some axes start at 0 others 500 min.

S3d/e traces are not labelled or colored to distinguish cells

Fig 6a the labels are too small. It is nice to show the entire ICE, but the relevant regions should be expanded so gene names are eligible, especially those discussed in the document.

6b please explain the bisR fold change compared to wt, when the bisR gene is deleted.

6d Please explain why there are two copies of InrR? P12,3-6. The second copy is not shown in Fig 1.

Fig 7 lacks detail and logic. The goal is to understand the function of the tc regulon genes (P12;15-21). These experiments do not address that question in a meaningful way. What is surprising is that substantial deletions of multiple regulon genes has no impact on conjugation. Why were these regions deleted in particular? Why were such large deletions created, rather than promoter deletions? Did the deletions include the promoters or just the genes? virD4 is essential for transfer, so its deletion along with other genes provides no insight on the function of the other genes; that deletion was always going to be transfer defective.

General comments

The word usage and syntax need addressing. I understand that English is not the native language; but the text would benefit from some editing.

This is a very complicated series of experiments involving many different genes and promoters, which made the work extremely difficult to follow. There are a-r gene segments, containing different promoters, with confusing 5-digit labels and names in small subscript . Fig 2 continues to use both the letter and gene assignment, but I don’t think this helps without having fig 1 on display. The lineups of the individual charts are different in each fig 2-5, making it difficult to follow, especially when the promoter labels are in subscript. I am not sure there is a simple solution, but readability would be dramatically enhanced by dropping the multi-digit labels and using a standard flow for each figure.

**Have all data underlying the figures and results presented in the manuscript been provided?**

Reviewer #1: Yes

Reviewer #2: Yes

Reviewer #3: Yes

PLOS authors have the option to publish the peer review history of their article (what does this mean?). If published, this will include your full peer review and any attached files.

Reviewer #1: No

Reviewer #2: No

Reviewer #3: No

---

## [Decision Letter · Decision Letter 1]

6 Jun 2022

Dear Dr van der Meer,

Thank you very much for submitting your Research Article entitled 'A bistable prokaryotic differentiation system underlying development of conjugative transfer competence' to PLOS Genetics.

The manuscript was fully evaluated at the editorial level and by independent peer reviewers. The reviewers 1 & 2 consider that you have addressed all their comments. However, reviewer 3 has still some concerns that we ask you address in the revised manuscript: This reviewer has previously asked a question considering the impact of large deletions of three regions shown in the Fig 7, and thinks that you have not addressed the issue in the revised manuscript. This reviewer considers that Fig and text are essentially the same and are misleading (p13 and 16, 1-12). One region is required for transfer. But it includes the known transfer gene virB4, so the role of the other ~10 genes cannot be interpreted. On P16, you ignore this fact and suggest that P81655 transcribes the second half of the operon. There is no evidence for this and the authors cannot rule out there are downstream internal promoters driving virB4 expression. The other region deleted is also not required for transfer (fig 7). Yet you continue to speculate that these large operons are “functionally important for ICE regulation and/or transfer???”(P16 10-12). And, if they are really part of the regulon that primes cells for transfer, how can their deletion have no impact on transfer? There is no interpretation of how these surprising results support their model.Reviewer 3 also finds that you have identified a potential consensus sequence (S3) but that you did not provide information where the motif maps relative to transcription and/or the gene. This reviewer considers that, as presented, this is not convincing, because the numbers indicate very different distances to the downstream gene.Finally, reviewer 3 considers that there is a lack of uniformity in text and figures for TraI and 50240, which are the same gene! Same for Int and IntB13, and InrR and InR. Fig S1 promoter fragments are all in the opposite orientation to Fig 1.

[LINK]

Yours sincerely,

Ivan Matic

Associate Editor

PLOS Genetics

Lotte Søgaard-Andersen

Section Editor: Prokaryotic Genetics

PLOS Genetics

Reviewer's Responses to Questions

**Comments to the Authors:**

Reviewer #1: The authors have addressed all my previous comments and I have no further recommendations. I think the paper is suitable for publication.

Reviewer #2: The authors have addressed virtually all of the reviewers' comments. I commend them on the thoroughness of the revisions and responses.

Reviewer #3: I still have some important points that need to be addressed:

1. Fig 7 shows the impact of large deletions of three regions. I queried this before and the authors have not addressed the issue. Fig and text are essentially the same and are misleading (p13 and 16, 1-12). One region is required for transfer. But it includes the known transfer gene virB4, so the role of the other ~10 genes cannot be interpreted. On P16, the authors ignore this fact and suggest that P81655 transcribes the second half of the operon. There is no evidence for this and the authors cannot rule out there are downstream internal promoters driving virB4 expression. The other region deleted is also not required for transfer (fig 7). Yet they continue to speculate that these large operons are “functionally important for ICE regulation and/or transfer???”(P16 10-12). And, if they are really part of the regulon that primes cells for transfer, how can their deletion have no impact on transfer? There is no interpretation of how these surprising results support their model.

2. They identify a potential consensus sequence (S3) but we are not told where the motif maps relative to transcription and/or the gene. As presented this is not convincing, especially as the numbers indicate very different distances to the downstream gene(?).

3. Although writing is improved the text is still not clear and word usage increases the difficulty of interpretation. Even the abstract has missing words and poor sentence construction. More care is needed in the text and figs to avoid making me concerned about other points I have missed and their overall rigor – also suggested by their data interpretation. Eg lack of uniformity in text and figures for TraI and 50240, which I realized eventually are the same gene! Same for Int and IntB13, and InrR and InR. Fig S1 promoter fragments are all in the opposite orientation to Fig 1.

**Have all data underlying the figures and results presented in the manuscript been provided?**

Reviewer #1: Yes

Reviewer #2: Yes

Reviewer #3: Yes

PLOS authors have the option to publish the peer review history of their article (what does this mean?). If published, this will include your full peer review and any attached files.

Reviewer #1: No

Reviewer #2: No

Reviewer #3: No

---

## [Editor Report · Decision Letter 2]

8 Jun 2022

Dear Dr van der Meer,

We are pleased to inform you that your manuscript entitled "A bistable prokaryotic differentiation system underlying development of conjugative transfer competence" has been editorially accepted for publication in PLOS Genetics. Congratulations!

Yours sincerely,

Ivan Matic

Associate Editor

PLOS Genetics

Lotte Søgaard-Andersen

Section Editor: Prokaryotic Genetics

PLOS Genetics

Comments from the reviewers (if applicable):

**Data Deposition**

http://datadryad.org/submit?journalID=pgenetics&manu=PGENETICS-D-21-01627R2

**Press Queries**

---

## [Editor Report · Acceptance letter]

22 Jun 2022

PGENETICS-D-21-01627R2 

A bistable prokaryotic differentiation system underlying development of conjugative transfer competence 

Dear Dr van der Meer, 

We are pleased to inform you that your manuscript entitled "A bistable prokaryotic differentiation system underlying development of conjugative transfer competence" has been formally accepted for publication in PLOS Genetics! Your manuscript is now with our production department and you will be notified of the publication date in due course.

With kind regards,

Olena Szabo

PLOS Genetics

On behalf of:
